# Adversarial Self Flow Matching: Few-steps Image Generation with Straight Flows

## Abstract

Flow Matching provides a method to train Ordinary Differential Equation (ODE)-based generative models and facilitates various probability path designs between initial and target distributions. Among these designs, straight flows are particularly interesting for reducing sampling steps. While some works have successfully straightened flows and achieved image generation in a few steps, they often suffer from cumulative errors or provide only piecewise or minibatch-level straightness. We propose Adversarial Self Flow Matching (ASFM), which can straighten flows and align the generated data distribution with the real data distribution. ASFM consists of two complementary components. Online Self Training straightens flows by constructing a conditional vector field using paired data, enabling one-step image generation during training. Adversarial Training aligns the one-step generated data with real data, thereby reducing cumulative errors when straightening flows. Experiments demonstrate that ASFM can build straight flows across the entire time span between two complete distributions and achieve highly competitive results across multiple datasets among Flow Matching-based methods. For instance, ASFM achieves 8.15 and 14.9 FID scores with NFE=6 on CelebA-HQ (256) and AFHQ-Cat (256), respectively.

## 1 Introduction

Flow Matching (FM) provides a straightforward method for obtaining ODE-based generative models by directly regressing on the conditional vector field. Unlike Diffusion Models (DMs), which constrain the probability path between distributions to follow a noise-adding and denoising process, FM offers a much broader design space for probability path. This simplicity and flexibility have attracted widespread attention across different fields, like image generation (Esser et al., 2024; Sun et al., 2024), structural biology (Huguet et al., 2024; Nori & Jin, 2024), audio generation (Wang et al., 2024) and optimal transport (Tong et al., 2023; Pooladian et al., 2023).

ODE-based generative models suffer from long sampling time. Although using more precise solvers can generate high-quality data in fewer steps, the core issue lies in the high curvature of the flows generated by the model (Lee et al., 2023). Given that FM offers a large design space for probability paths, finding models within this space that can generate straight flows becomes the key to accelerating sampling in ODE-based generative models. Some recent works have already attempted to straighten the flows. Liu et al. (2022) suggest using the Reflow method to reduce the intersections of straight line paths, thereby straightening the sampling trajectories. However, since generated data is used as the training set during the Reflow training process, an accumulation of errors becomes inevitable, causing the final generated data distribution to deviate from the real data distribution. The methods proposed in Yan et al. (2024) and Yang et al. (2024) for straightening flows require segmenting the flow in the time dimension, resulting in piecewise straight flows. However, they do not guarantee that the flow is straight over the entire time span. Both Tong et al. (2023) and Pooladian et al. (2023) propose non-trivial pairing of noise and data samples within a minibatch during training to straighten the flow. Nevertheless, these methods fail to achieve optimal pairing for data outside the minibatch, making it impossible to construct straight flows between two complete distributions.

To enable the model to generate straight flows between two complete distributions over the entire time span, while ensuring that the generated data does not deviate from the real data distribution,

we propose Adversarial Self Flow Matching (ASFM), and summarize it in Figure 1[1]. ASFM is primarily divided into two components: Adversarial Training and Online Self Training. Adversarial Training leverages the straight flow characteristic of ASFM to obtain the generated data in a single step during training. This result, along with real data, is fed into the discriminator, ensuring that the distribution of generated data closely aligns with the real data distribution. On the other hand, Online Self Training generates training data using the model itself, progressively straightening the flow by learning the vector field with repaired data, ensuring that the flow remains sufficiently straight over the entire time span and between two complete distributions. These two components complement each other: Adversarial Training ensures that the training data generated by Online Self Training aligns better with the real data distribution, while Online Self Training straightens the flow and thereby improves the quality of one-step generation in Adversarial Training.

Our contributions can be summarized as follows:

1. Our method demonstrates that within the broad probability path design space provided by Flow Matching, finding straight flows between two complete high-dimensional distributions across the entire time span is feasible. ASFM finds such flows using complementary Adversarial Training and Online Self Training, and fundamentally reduces the number of steps required for sampling in ODE-based generative models.

2. We introduce Adversarial Training into the Flow Matching training process, using a discriminator to align the distribution of generated data with that of real data, which reduces the cumulative errors in the Online Self Training process. ASFM provides new possibilities for integrating adversarial training with ODE-based generative models.

3. We conducted extensive experiments to validate the effectiveness of our method. On CIFAR-10, ASFM can achieve FID scores of 5.07 and 4.56 with NFE=1 and NFE=2, respectively. On high-resolution datasets like AFHQ-Cat and CelebA-HQ, ASFM achieves FID scores of 14.9 and 8.15 with NFE=6, respectively.

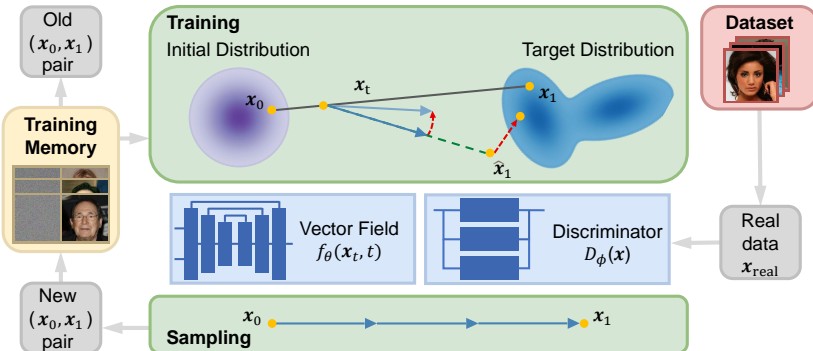

Figure 1: The pipeline of ASFM. Training Memory stores paired data $(\boldsymbol{x}_0, \boldsymbol{x}_1)$ for training, continually updated by discarding old pairs and adding newly generated pairs. During training, vector field $f$ is trained using Online Self Training and Adversarial Training. The former pulls the vector field to align with the direction from $\boldsymbol{x}_0$ to $\boldsymbol{x}_1$ ($\boldsymbol{x}_0$ and $\boldsymbol{x}_1$ are repaired), while the latter draws the one-step generated data $\hat{\boldsymbol{x}}_1$ closer to the distribution of real data. Additionally, discriminator $D$ is trained using both real data $\boldsymbol{x}_{real}$ and $\hat{\boldsymbol{x}}_1$. During sampling, ODE solvers (such as Euler) are used to generate data.

## 2 RELATED WORKS

**Flow Matching** Training ODE-based generative models through regression is not feasible, as the ground-truth vector field $u_t(\boldsymbol{x})$ cannot be directly obtained. Flow Matching (FM) (Lipman et al., 2022) proposes regressing on the conditional vector field $u_t(\boldsymbol{x}|\boldsymbol{x}_1)$ to solve this problem. Similarly,

---

[1]To keep things concise, we use a single $\boldsymbol{x}_t$ to illustrate both Online Self Training and Adversarial Training.

Liu et al. (2022) proposes learning the vector field to follow the straight paths between samples from two distributions, while Albergo & Vanden-Eijnden (2022) suggests randomly interpolating between sampling points of two distributions to learn the vector field. These works share similar principles and offer new, simplified methods for training ODE-based generative models. Unlike Diffusion Models (DMs), which constrain probability paths to follow a noise-adding and denoising process, these approaches allow for greater flexibility in designing probability paths. In this case, designing ODE-based generative models that generate straight flows becomes an attractive task, as straight flows can fundamentally accelerate the sampling process. Yang et al. (2024) enforces self-consistency property in the vector field, enabling the flow to exhibit piecewise straight characteristics. Yan et al. (2024) divides the flow into multiple time segments and applies Reflow (Liu et al., 2022) within each segment, transforming the flow into a piecewise straight one. Pooladian et al. (2023) and Tong et al. (2023) straighten the flow within a minibatch by pairing the sampling points from the two distributions within the minibatch. Additionally, Lee et al. (2023) reduces the trajectory curvature of ODE-based generative models by learning the forward process.

**Adversarial Training** Generative Adversarial Networks (GANs) (Goodfellow et al., 2020; Sauer et al., 2022b; 2023a) use adversarial training to improve the quality of generated images: the generator tries to create images that can deceive the discriminator, while the discriminator attempts to determine whether the images come from the real image dataset. Some recent works have attempted to apply adversarial training to Stochastic Differential Equation (SDE)/ODE-based generative models, but these efforts have primarily focused on DMs and distillation-related tasks. The application of adversarial training in FM is still relatively unexplored. Kong et al. (2024) uses Adversarial Consistency Training to reduce the Jensen-Shannon divergence between distributions, improving both generation quality and convergence of DMs. Luo et al. (2024); Xiao et al. (2021); Gong et al. (2024); Xu et al. (2024) propose several diffusion-GAN hybrid models that aim to leverage the strengths of both approaches while mitigating their weaknesses, resulting in models with faster sampling speeds and better mode coverage. Sauer et al. (2023b) applies adversarial training in model distillation, where the discriminator distinguishes between the student model's output and real images to improve generation quality. Lee et al. (2024) applies adversarial training to flow matching in the field of waveform generation. It is important to note that, not only is this work in a different domain from ours, but the algorithms are also significantly different. Our method leverages the mutual enhancement between Online Self Training and Adversarial Training, continuously straightening the flow while ensuring that the distribution of generated data aligns with that of real data. In contrast, their method does not incorporate any online self training components.

## 3 METHOD

### 3.1 PRELIMINARIES: FLOW MATCHING AND ADVERSARIAL TRAINING

Let $p_0$ and $p_1$ denote initial distribution and target distribution on data space $\mathbb{R}^n$, respectively. An ODE-based generative model aims to transform sampling point $\boldsymbol{x}_0$ from the initial distribution into some sampling point $\boldsymbol{x}_1$ from the target distribution, and the dynamics of the sampling point are described by the following ODE:

$$\frac{d}{d\mathrm{t}}\Gamma_\mathrm{t}(\boldsymbol{x}) = f(\boldsymbol{x}, \mathrm{t}; \boldsymbol{\theta}), \tag{1}$$

$$\Gamma_0(\boldsymbol{x}) = \boldsymbol{x}_0, \tag{2}$$

where $\mathrm{t} \in [0, 1]$. In the above equation, $f(\boldsymbol{x}, \mathrm{t}; \boldsymbol{\theta}) : \mathbb{R}^n \times [0, 1] \to \mathbb{R}^n$ represents a *time-dependent vector field*, and $\Gamma_\mathrm{t}(\boldsymbol{x})$ is a *flow*[2] that describes the trajectory of sample point $\boldsymbol{x}_\mathrm{t}$ starting from $\boldsymbol{x}_0$ at $\mathrm{t} = 0$. Besides, a *probability density path*[3] is a time-dependent probability density function $p(\boldsymbol{x}, \mathrm{t}) : \mathbb{R}^n \times [0, 1] \to \mathbb{R}^n$ describing how the distribution of $\boldsymbol{x}_\mathrm{t}$ changes over time t while satisfying $\int p_\mathrm{t}(\boldsymbol{x})d\boldsymbol{x} = 1$. The vector field and the probability path are related through the following *continuity equation* (Villani et al., 2009):

$$\partial_\mathrm{t} p_\mathrm{t}(\boldsymbol{x}) = -\nabla \cdot (f(\boldsymbol{x}, \mathrm{t})p_\mathrm{t}(\boldsymbol{x})). \tag{3}$$

---

[2]For simplicity, in this paper we will sometimes use time t as a subscript. For example, we use $\Gamma_\mathrm{t}(\boldsymbol{x})$ to represent $\Gamma(\boldsymbol{x}, \mathrm{t})$, $p_\mathrm{t}(\boldsymbol{x})$ to represent $p(\boldsymbol{x}, \mathrm{t})$ and $f_\mathrm{t}(\boldsymbol{x})$ to represent $f(\boldsymbol{x}, \mathrm{t})$.

[3]We will also refer to the probability density path as the probability path for simplicity.

FM (Lipman et al., 2022) represents the vector field using a neural network, with parameters $\boldsymbol{\theta}$. An intuitive optimization objective $\mathcal{L}_{FM}(\boldsymbol{\theta})$ for training this network is:

$$\mathbb{E}_{t, p_t(\boldsymbol{x})}[||f_t(\boldsymbol{x}) - u_t(\boldsymbol{x})||^2], \tag{4}$$

where $u_t$ is the ground-truth vector field. However, learning vector field $f_t(\boldsymbol{x})$ from $u_t(\boldsymbol{x})$ is infeasible since $u_t(\boldsymbol{x})$ is difficult to obtain directly. FM provides a novel way to solve this problem: it allows $f_t(\boldsymbol{x})$ to regress on the conditional vector field $u_t(\boldsymbol{x}|\boldsymbol{x}_1)$ and proves that Equation 4 and the following optimization objective $\mathcal{L}_{CFM}$ provide the same gradient information for $\boldsymbol{\theta}$:

$$\mathbb{E}_{t, q(\boldsymbol{x}_1), p_t(\boldsymbol{x}|\boldsymbol{x}_1)}[||f_t(\boldsymbol{x}) - u_t(\boldsymbol{x}|\boldsymbol{x}_1)||^2], \tag{5}$$

where $q$ is the distribution of real data.

The core mechanism of adversarial training is to have the generator and discriminator compete: the generator tries to create data that can confuse the discriminator, while the discriminator attempts to distinguish the generated data from real data (Goodfellow et al., 2020). It can be expressed as the following minimax optimization problem:

$$\min_{G} \max_{D} \mathcal{V}(\boldsymbol{\phi}, \boldsymbol{\psi}) = \mathbb{E}_{\boldsymbol{x}_{real} \sim p_1}[D(\boldsymbol{x}_{real}; \boldsymbol{\phi})] + \mathbb{E}_{\boldsymbol{x}_0 \sim p_0}[-D((G(\boldsymbol{x}_0; \boldsymbol{\psi})); \boldsymbol{\phi})], \tag{6}$$

where $D(\boldsymbol{x}; \boldsymbol{\phi})$ represents the discriminator's prediction and $G(\boldsymbol{x}_0; \boldsymbol{\psi})$ is the data generated by the generator.

## 3.2 CONSTRUCTING STRAIGHT FLOWS POINTING TOWARD REAL DATA DISTRIBUTION

Given a set of initial random samples, our task is to construct straight flows that progressively transform these samples to align with the distribution of real data. Inspired by recent methods (Lipman et al., 2022; Liu et al., 2022; Sauer et al., 2023b), we continuously straighten flows through Online Self Training and use Adversarial Training to guide the generated data to align with the real data distribution, creating a novel ODE-based generative model based on the Flow Matching method. Our innovation lies in establishing a training framework where the model continuously generates data for training itself, while incorporating Adversarial Training into FM. The overview of the proposed method is illustrated in Figure 1. In Section 3.2.1, we introduce Online Self Training, which is implemented using two components: Flow Matching with paired data and Training Memory. Then Section 3.2.2 discusses how to perform one-step generation during training and apply the generated results in adversarial training. Section 3.2.3 summarizes the algorithm and presents the complete training procedure.

### 3.2.1 ONLINE SELF TRAINING

Online Self Training means that the model generates data to train itself (self training), and after updating the model parameters, it generates new data for training (online training), continuously repeating this process. Inspired by Reflow (Liu et al., 2022), in order to obtain straight flows, we can train the model using paired data generated by the model itself. This leverages the non-crossing property of flows to repair data and therefore continuously reduces the occurrence of linear interpolation intersections between training pairs, resulting in straight flows. As shown in Figure 2, when the linear interpolations of two sets of paired data $(\boldsymbol{x}_0, \boldsymbol{x}_1)$ and $(\boldsymbol{x}_0', \boldsymbol{x}_1')$ intersect at certain time t, the learned vector field $f_t$ will reassign their pairing. As the new pairings $(\boldsymbol{x}_0, \boldsymbol{x}_1')$ and $(\boldsymbol{x}_0', \boldsymbol{x}_1)$ are used as training

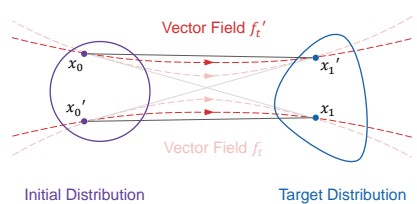

Figure 2: Model-generated training data leads to repairing, and using the new pairs to train the vector field $f_t$ will result in a straighter, updated vector field $f_t'$.

data, no intersections exist between their linear interpolations. Therefore, the flows guided by the newly learned vector field $f_t'$ no longer need to approach intersection points. Given that the sum of the lengths of two sides of a triangle is greater than the third side, and the straight line between two points is the shortest path, the learned vector field $f_t'$ will become straighter. It should be emphasized

that how ReFlow straightens the vector field is incomplete. Specifically, an $(m + 1)$-rectified flow can only try to eliminate intersections that occur in an $m$-rectified flow, but cannot remove intersections that may exist within itself. In contrast, Online Self Training continuously updates the model, reducing the likelihood of creating linear interpolation intersections in generated data pairs in real time. Additionally, our method is more efficient in utilizing data, which we will elaborate in detail through experiments in Section 4.3.

**Flow Matching with Paired Data** To train model with paired data, we extend the conditional flow matching to a case conditioned on paired data, which can be expressed by the following equation,

$$\mathbb{E}_{t, q(\boldsymbol{x}_0, \boldsymbol{x}_1), p_t(\boldsymbol{x}|\boldsymbol{x}_0, \boldsymbol{x}_1)}[||f_t(\boldsymbol{x}) - u_t(\boldsymbol{x}|\boldsymbol{x}_0, \boldsymbol{x}_1)||^2], \tag{7}$$

where $q$ represents the joint distribution of $\boldsymbol{x}_0$ and $\boldsymbol{x}_1$. And when flow (conditioned on $\boldsymbol{x}_0$ and $\boldsymbol{x}_1$) can be described as $\Gamma_t(\boldsymbol{x}) = (1 - t)\boldsymbol{x}_0 + t\boldsymbol{x}_1$, $t \in [0, 1]$, Equation 7 can be written as

$$\mathbb{E}_{t, q(\boldsymbol{x}_0, \boldsymbol{x}_1), p_t(\boldsymbol{x}|\boldsymbol{x}_0, \boldsymbol{x}_1)}[||f_t(\boldsymbol{x}) - (\boldsymbol{x}_1 - \boldsymbol{x}_0)||^2]. \tag{8}$$

Denote Equation 5 as $\mathcal{L}_{CFM}(\boldsymbol{\theta})$ and Equation 7 as $\mathcal{L}_{CFM+}(\boldsymbol{\theta})$. We propose the following proposition. For the detailed proof, please refer to Appendix A.

**Proposition 1.** *Assuming that $p_t(\boldsymbol{x}) > 0$ for all $\boldsymbol{x} \in \mathbb{R}^n$ and $t \in [0, 1]$, then, $\mathcal{L}_{CFM+}$ and $\mathcal{L}_{CFM}$ are equal, up to a constant independent of $\boldsymbol{\theta}$. Therefore, $\nabla_{\boldsymbol{\theta}}\mathcal{L}_{CFM+}(\boldsymbol{\theta}) = \nabla_{\boldsymbol{\theta}}\mathcal{L}_{FM}(\boldsymbol{\theta}) = \nabla_{\boldsymbol{\theta}}\mathcal{L}_{CFM}(\boldsymbol{\theta})$. And when flow (conditioned on $\boldsymbol{x}_0$ and $\boldsymbol{x}_1$) can be described as $\Gamma_t(\boldsymbol{x}) = (1 - t)\boldsymbol{x}_0 + t\boldsymbol{x}_1$, $\mathcal{L}_{CFM+}$ is the same as Equation 8.*

**Training Memory** During our experiments, we found that when using Online Self Training directly, where only the newly generated data is used each time the model is trained, causing the model's training to fail to converge, as demonstrated in the ablation study of Section 4.3. To address this issue, we propose a Training Memory module. This module stores a fixed amount of paired training data, for example $k$ pairs of $(\boldsymbol{x}_0, \boldsymbol{x}_1)$. Newly generated training pairs are added to this Training Memory, and when the pairs exceed the module's storage capacity, the oldest pairs are discarded in a first-in-first-out manner. When selecting pairs for the current training, A batch of pairs is randomly sampled with replacement from the Training Memory. Thus, when the Training Memory has not been updated, the optimization objective for training is as follows:

$$\mathbb{E}_{t, q_{TM}(\boldsymbol{x}_0, \boldsymbol{x}_1), p_t(\boldsymbol{x}|\boldsymbol{x}_0, \boldsymbol{x}_1)}[||f_t(\boldsymbol{x}) - u_t(\boldsymbol{x}|\boldsymbol{x}_0, \boldsymbol{x}_1)||^2], \tag{9}$$

where $q_{TM}(\boldsymbol{x}_0, \boldsymbol{x}_1)$ represents the joint distribution of $\boldsymbol{x}_0$ and $\boldsymbol{x}_1$ from Training Memory. This design retains the real-time updating characteristic of online training while making the training process more stable.

### 3.2.2 ADVERSARIAL TRAINING

When using ODE numerical solution methods like Euler to generate data, as shown in Figure 3, errors are introduced. Additionally, since our method adopts an Online Self Training approach, training the model with generated data that contains certain deviations can lead to the accumulation of these errors. Eventually, this will cause the generated data distribution to deviate from the real data distribution. To solve this problem, we incorporate Adversarial Training into our method to make the generated data distribution align more closely with the real data distribution as shown in Figure 3.

**One-step Generation in Training Process** Adversarial Training requires obtaining generated data during training. However, unlike the generator in GANs, which can generate data with just one evaluation of the network, ODE-based generative models require the use of an ODE solver and multiple evaluations of the network. This not only

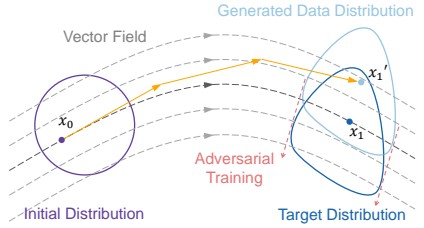

Figure 3: $\boldsymbol{x}_1$ and $\boldsymbol{x}_1'$ are respectively the results of moving along the vector field and the numerical solution at t = 1, starting from $\boldsymbol{x}_0$. The difference between $\boldsymbol{x}_1$ and $\boldsymbol{x}_1'$ reflects the deviation of the generated data distribution from the target distribution, and Adversarial Training attempts to eliminate this deviation.

significantly increases the time needed for each step of training but also causes a substantial increase in size of gradient information, leading to higher memory usage. To overcome this issue, our method leverages the sufficiently straight sampling trajectories of the model and generates data in one-step during training by:

$$G(\boldsymbol{x}_\mathrm{t}, \mathrm{t}) = \hat{\boldsymbol{x}}_1 = \boldsymbol{x}_\mathrm{t} + (1 - \mathrm{t})f(\boldsymbol{x}_\mathrm{t}, \mathrm{t}), \tag{10}$$

where $\boldsymbol{x}_\mathrm{t}$ is a sample point of $p_t(\boldsymbol{x}|\boldsymbol{x}_0, \boldsymbol{x}_1)$. Notably, the above equation holds under the condition that the sampling trajectories are sufficiently straight. Then, denoting discriminator as $D$ and its parameters as $\boldsymbol{\phi}$, we can incorporate the adversarial loss $\mathcal{L}_{Adv}(\boldsymbol{\theta})$ when training vector field $f_\mathrm{t}$:

$$D(\hat{\boldsymbol{x}}_1; \boldsymbol{\phi}) = D(G(\boldsymbol{x}_\mathrm{t}, \mathrm{t}); \boldsymbol{\phi}). \tag{11}$$

When training discriminator $D$, we use the one-step generated data $\hat{\boldsymbol{x}}_1$ along with real data $\boldsymbol{x}_{real}$ to get the discriminator loss $\mathcal{L}_{Disc}(\boldsymbol{\phi})$:

$$\mathcal{L}_{Disc}(\boldsymbol{\phi}) = D(\hat{\boldsymbol{x}}_1) - D(\boldsymbol{x}_{real}) = D(G(\boldsymbol{x}_\mathrm{t}, \mathrm{t})) - D(\boldsymbol{x}_{real}). \tag{12}$$

In this way, Online Self Training and Adversarial Training complement each other: on the one hand, Online Self Training causes the flows produced by the model to become increasingly straight, ensuring that Adversarial Training can obtain accurate $\hat{\boldsymbol{x}}_1$ through one-step generation during the training process. On the other hand, Adversarial Training ensures that the vector field at any point $\boldsymbol{x}_\mathrm{t}$ points toward the manifold of the real data, reducing the cumulative error that may arise from the data generation process in Online Self Training.

### 3.2.3 TRAINING PROCEDURE

In this section, we summarize the entire training procedure. At the beginning of training, each training step generates data to add to the Training Memory until it is filled up. Afterward, new training data is generated and added to update Training Memory at intervals determined by the update frequency $freq$. This ensures that the data in the Training Memory is fully utilized across all time t. When updating the parameters, the vector field network parameters $\boldsymbol{\theta}$ are updated according to the following loss,

$$\begin{aligned} Loss_1(\boldsymbol{\theta}) \\ = \mathcal{L}_{CFM+}(\boldsymbol{\theta}) + \lambda_1 \mathcal{L}_{Adv}(\boldsymbol{\theta}) + \\ \lambda_2 \mathcal{L}_{Addition}(\boldsymbol{\theta}) \end{aligned} \tag{13}$$

where $\lambda_1$ is the weight coefficient for $\mathcal{L}_{Adv}$, $\mathcal{L}_{Addition}$ is an additional loss, such as the LPIPS loss (Zhang et al., 2018) that can enhance the visual quality of generated images and $\lambda_2$ is its weight coefficient. Network parameters $\phi$ of the discriminator are updated based on the following loss,

$$Loss_2(\boldsymbol{\phi}) = \mathcal{L}_{Disc}(\boldsymbol{\phi}). \tag{14}$$

---

**Algorithm 1:** Training

**Input:** Vector field $f(\boldsymbol{x}, \mathrm{t}; \boldsymbol{\theta})$, Discriminator $D(\boldsymbol{x}; \boldsymbol{\phi})$, Training Memory $Q$, Training Memory size $k$, Training Memory update frequency $freq$, maximum training steps $max\_step$, ODE solver $Solver(\boldsymbol{x}_0, f_\mathrm{t})$, learning rate $\eta_1$ and $\eta_2$

Initialize Training Memory $Q = []$;
**for** $step = 0$ *to* $max\_step$ **do**
  **if** $step < k$ *or* $step \bmod freq == 0$
    Sample $\boldsymbol{x}_0$ from $\mathcal{N}(0, 1)$;
    $\boldsymbol{x}_1 = Solver(\boldsymbol{x}_0, f(\boldsymbol{x}_\mathrm{t}, \mathrm{t}; \boldsymbol{\theta}^-))$;
    $Q$.enqueue$((\boldsymbol{x}_0, \boldsymbol{x}_1))$;
  **if** $len(Q) > k$
    $Q$.dequeue() ;  *// Online Self Training*
  Sample $(\tilde{\boldsymbol{x}}_0, \tilde{\boldsymbol{x}}_1)$ from $Q$;
  Sample $\mathrm{t}_1$ from $U[0, 1]$;
  $\tilde{\boldsymbol{x}}_{\mathrm{t}_1} = (1 - \mathrm{t}_1)\tilde{\boldsymbol{x}}_0 + \mathrm{t}_1\tilde{\boldsymbol{x}}_1$;
  $\hat{\boldsymbol{x}}_1 = G(\tilde{\boldsymbol{x}}_{\mathrm{t}_1}, \mathrm{t}_1)$;
  $Loss_1 = \|f(\tilde{\boldsymbol{x}}_{\mathrm{t}_1}, \mathrm{t}_1; \boldsymbol{\theta}) - (\tilde{\boldsymbol{x}}_1 - \tilde{\boldsymbol{x}}_0)\|^2 +$
    $\lambda_1 D(\hat{\boldsymbol{x}}_1; \boldsymbol{\phi}) + \lambda_2 lpips(\hat{\boldsymbol{x}}_1, \tilde{\boldsymbol{x}}_1)$;
  $\boldsymbol{\theta} = \boldsymbol{\theta} - \eta_1 \nabla Loss_1$;
  Sample $\mathrm{t}_2$ from $U[0, 1]$;
  $\tilde{\boldsymbol{x}}_{\mathrm{t}_2} = (1 - \mathrm{t}_2)\tilde{\boldsymbol{x}}_0 + \mathrm{t}_2\tilde{\boldsymbol{x}}_1$;
  $\hat{\boldsymbol{x}}_1' = G(\tilde{\boldsymbol{x}}_{\mathrm{t}_2}, \mathrm{t}_2)$;
  $Loss_2 = D(\hat{\boldsymbol{x}}_1'; \boldsymbol{\phi}) - D(\boldsymbol{x}_{real}; \boldsymbol{\phi})$;
  $\boldsymbol{\phi} = \boldsymbol{\phi} - \eta_2 \nabla Loss_2$;  *// Adversarial Training*

---

We summarize the entire training process in Algorithm 1. Parameters of the neural network that are processed by Exponential Moving Average (EMA), are marked with a superscript $-$, i.e., $\boldsymbol{\theta}^-$. For specific parameter settings, please refer to Appendix B.

# 4 EXPERIMENTS

**Experimental Settings** We validated the effectiveness of our method on the task of unconditional image generation using three datasets. Specifically, we used the CIFAR-10 (Krizhevsky et al., 2009) dataset with $32 \times 32$ images in Section 4.1, and two high-resolution datasets with $256 \times 256$ images, i.e., CelebA-HQ (Karras, 2017) and AFHQ-Cat (Choi et al., 2020) in Section 4.2. For the evaluation metrics, we selected the Frechet Inception Distance (FID) score (Heusel et al., 2017) and Inception Score (IS) (Salimans et al., 2016), which are widely used in the field of image generation. The vector field network $f(\boldsymbol{x}, t; \boldsymbol{\theta})$ in our model is initialized using the checkpoint of the pretrained 1-rectified flow, and its architecture is the U-Net architecture of DDPM++ (Song et al., 2020b). The discriminator part $D$ uses the discriminator model from StyleGAN-T (Sauer et al., 2023a).

**Baselines** We compared ASFM with various methods on the CIFAR-10 dataset, including several typical GANs, Consistency Models, representative Diffusion Models (including some advanced solvers and distillation methods), and recent works related to FM. On high-resolution datasets, we mainly compared ASFM with Rectified Flow (Liu et al., 2022), Consistency Flow Matching (Yang et al., 2024), BOSS (Nguyen et al., 2023), and MTC (Lee et al., 2023). It is worth mentioning that although ASFM uses adversarial training, its theoretical foundation remains the same as ODE-based generative models, thus giving it zero-shot capabilities in downstream tasks that GANs cannot achieve. For detailed zero-shot experiments, please refer to Appendix C.

## 4.1 ASFM IS HIGHLY COMPETITIVE AMONG FM-RELATED METHODS

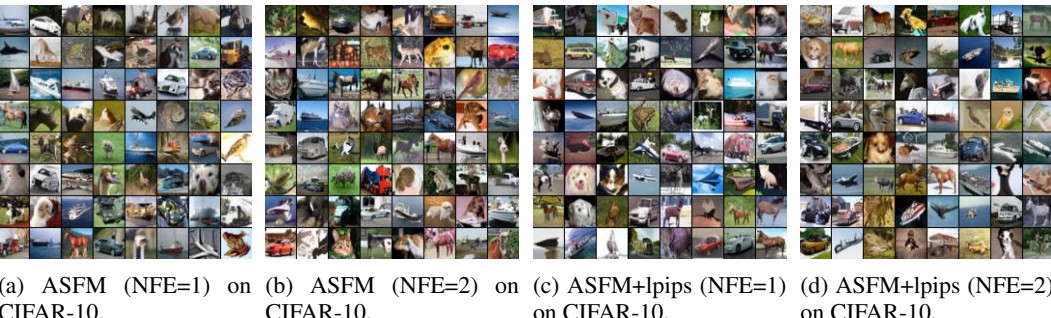

(a) ASFM (NFE=1) on CIFAR-10.  (b) ASFM (NFE=2) on CIFAR-10.  (c) ASFM+lpips (NFE=1) on CIFAR-10.  (d) ASFM+lpips (NFE=2) on CIFAR-10.

Figure 4: Generated images by ASFM, sampling with Euler Solver in 1 or 2 steps. The two images on the right are generated by ASFM trained with a loss function that includes the LPIPS loss.

The generative performance of ASFM on CIFAR-10 dataset is shown in Figure 4 and quantitative results are given in Table 1. As seen in Table 1, ASFM achieves competitive results in unconditional image generation on CIFAR-10. With Number of Function Evaluations (NFE)=1 and NFE=2, the FID scores reach 5.65 and 4.90, respectively. When the LPIPS loss is added to the loss function, the FID scores reach 5.07 and 4.56 with NFE=1 and NFE=2, surpassing those of several representative GANs, DMs, DMs+Advanced Solvers, and DMs+Distillation. Additionally, the performance of ASFM+lpips with NFE=2 exceeds all the FM-related works with same or more NFE.

## 4.2 HIGH-RESOLUTION IMAGE GENERATION WITH EXTREMELY FEW STEPS

Table 2 presents the quantitative results of ASFM and other FM-related methods on high-resolution datasets, and generated images are given in Figure 5. It is evident that even with NFE=1, ASFM outperformed 1-rectified flow with NFE=8. Moreover, ASFM with NFE=2 achieves results comparable to recent works like Consistency FM and BOSS with NFE=6. And when the NFE of ASFM reaches 6, its performance significantly surpassed those of all other methods.

Table 1: Sample quality on CIFAR-10. Lower FID and higher IS are better.

| Method | NFE | FID (↓) | IS (↑) |
|---|---|---|---|
| **GAN** | | | |
| BigGAN (Brock, 2018) | 1 | 14.7 | **9.22** |
| AutoGAN (Gong et al., 2019) | 1 | 12.4 | 8.55 |
| TransGAN (Jiang et al., 2021) | 1 | 9.15 | 8.80 |
| StyleGAN2-ADA (Karras et al., 2020) | 1 | **2.39** | **10.0** |
| StyleGAN-XL (Sauer et al., 2022a) | 1 | **1.85** | - |
| **Diffusion** | | | |
| DDPM (Ho et al., 2020) | 1000 | 3.17 | **9.46** |
| DDIM (Song et al., 2020a) | 20 | 6.84 | - |
| DPM-Solver-2 (Lu et al., 2022) | 10 | 5.94 | - |
| Knowledge Distillation (Luhman & Luhman, 2021) | 1 | 9.36 | - |
| Progressive Distillation (Salimans & Ho, 2022) | 1 | 8.34 | 8.69 |
| Score SDE (Song et al., 2020b) | 2000 | **2.20** | 9.89 |
| EDM (Karras et al., 2022) | 35 | **1.97** | - |
| **CM** | | | |
| CT (Song et al., 2023) | 1 | **8.70** | **8.49** |
| CT | 2 | **5.83** | **8.85** |
| **FM** | | | |
| 1-Rectified Flow (Liu et al., 2022) | 1 | 379 | 1.13 |
| 2-Rectified Flow (Liu et al., 2022) | 1 | 12.2 | 8.13 |
| 3-Rectified Flow (Liu et al., 2022) | 1 | 7.43 | 8.56 |
| 1-Rectified Flow | 2 | 170 | 2.93 |
| 2-Rectified Flow | 2 | 6.34 | 8.69 |
| 3-Rectified Flow | 2 | 5.22 | 8.75 |
| OT-CFM (Tong et al., 2023) | 1 | 231 | - |
| OT-CFM | 2 | 93.4 | - |
| MTC (Lee et al., 2023) | 5 | 18.7 | 8.09 |
| MTC | 9 | 8.66 | 8.67 |
| Consistency FM (Yang et al., 2024) | 1 | 50.3 | 6.88 |
| Consistency FM | 2 | 5.38 | 8.67 |
| BOSS (Nguyen et al., 2023) | 6 | **4.80** | - |
| ASFM (Ours) | 1 | 5.80 | 8.89 |
| ASFM (Ours) | 2 | 4.82 | **8.93** |
| ASFM+lpips (Ours) | 1 | 5.07 | 8.85 |
| ASFM+lpips (Ours) | 2 | **4.56** | **9.02** |

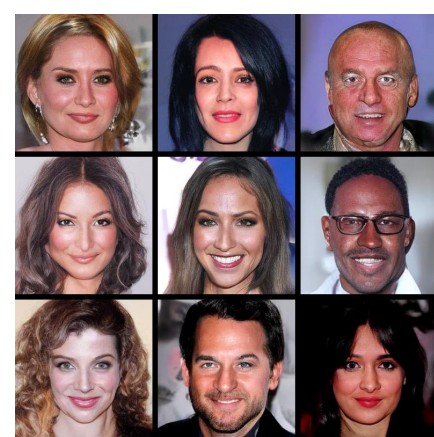

(a) ASFM (NFE=6) on AFHQ.          (b) ASFM (NFE=6) on CelebA-HQ.

Figure 5: Generated high-resolution images by ASFM, sampling with Euler Solver in 6 steps.

### 4.3 ABLATION STUDY

To validate the effectiveness of various designs in ASFM, we conduct ablation experiments on each module. We design an offline model as the baseline model, which generates 512 batches of $(\boldsymbol{x}_0, \boldsymbol{x}_1)$ pairs as a training set (batch size = 512). Then, we design three models:

A. Online Self Training + Training Memory of size 1 batch.

Table 2: Sample quality on high-resolution datasets, i.e., AFHQ and CelebA-HQ.

| Method | AFHQ-Cat 256 × 256 | | CelebA-HQ 256 × 256 | |
|---|---|---|---|---|
| | NFE | FID (↓) | NFE | FID (↓) |
| 1-Rectified-Flow (Liu et al., 2022) | 1 | 217 | 1 | 264 |
| 1-Rectified-Flow | 2 | 95.5 | 2 | 195 |
| 1-Rectified-Flow | 6 | 57.2 | 6 | 101 |
| 1-Rectified-Flow | 8 | 51.1 | 8 | 78.7 |
| Consistency FM (Yang et al., 2024) | 6 | 22.5 | 6 | 36.4 |
| BOSS (Nguyen et al., 2023) | 6 | 26.10 | 6 | 18.67 |
| MTC (Lee et al., 2023) | - | - | 128 | 10.4 |
| ASFM (Ours) | 1 | 44.0 | 1 | 79.0 |
| ASFM (Ours) | 2 | 22.9 | 2 | 22.2 |
| ASFM (Ours) | 6 | **14.9** | 6 | **8.15** |

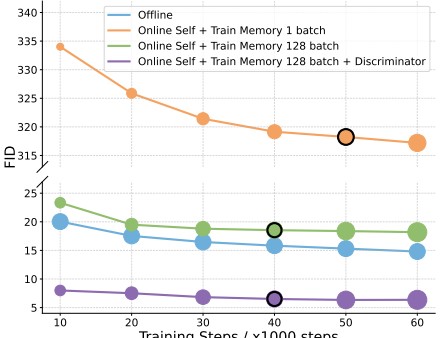

(a) Sampling with Euler Solver in 1 step.

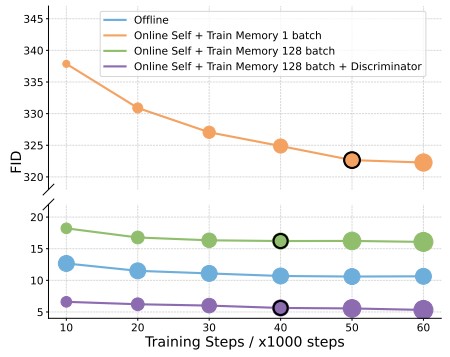

(b) Sampling with Euler Solver in 2 steps.

Figure 6: The image generation quality (measured by FID) of the Baseline Model and Models A, B, and C on CIFAR-10 as training steps progress is shown in (a) and (b). Both figures use the Euler Solver, but the former uses 1 sampling step, while the latter uses 2. The size of each data point in the figures represents the amount of data used at that training step, and the points marked with black borders indicate those using the same data amount as the Baseline Model.

    B. Online Self Training + Training Memory of size 128 batches.

    C. Online Self Training + Training Memory of size 128 batches + Adversarial Training.

Each model is trained for 60k steps and when using Online Self Training, the model generates 1 batch of data every 100 steps to update the Training Memory. Figure 6 shows the change in the quality of generated images for each model as training progresses. As can be seen from the figure, the FID of Model A remains high consistently, indicating that training using only newly generated data leads to failure in model convergence. However, when the size of Training Memory is increased to 128 batches, the model converges properly, demonstrating the effectiveness of Training Memory. Model C, which adds Adversarial Training on top of Model B, significantly improves image generation quality, surpassing the Baseline Model, thus proving the effectiveness of Adversarial Training.

Table 3: Comparison of image generation quality on CIFAR-10 between ASFM and 2-Rectified-Flow-Online at the same training step, using different amounts of data.

| Method | Training Steps | Data Size | FID (↓) | |
|---|---|---|---|---|
| | | | NFE = 1 | NFE = 2 |
| 2-Rectified-Flow-Online | 50000 | 50000×512 | 15.80 | 11.00 |
| ASFM (Ours) | 50000 | (500+127)×512 | **6.34** | **5.56** |

Although rectified flow offers an approach to train k-rectified flow (k ≥ 2) using Online Training, we need to point out that it does not employ Self Training. Its training data is generated by a fixed-parameter model, which is not directly related to the model being trained. In contrast, in our method, the model generating the training data is the same as the model being trained. Online Self Training

can adjust the model parameters in real-time to straighten the flows. A comparison between ASFM and the 2-Rectified-Flow trained using Online Training is shown in Table 3. From the table, it is clear that our model uses less data and achieves better generation results, demonstrating that ASFM has a higher data utilization efficiency.

## 5 CONCLUSIONS

We propose Adversarial Self Flow Matching (ASFM), a Flow Matching (FM)-based method that generates high-quality images in a few steps by leveraging straightened flows. Our method employs Online Self Training to straighten flows, where intersections between the linear interpolations of training pairs are eliminated in real time. Furthermore, our approach incorporates Adversarial Training to reduce the deviation between the generated data distribution and the real data distribution caused by ODE solvers. Experiments demonstrate that ASFM can generate straight flows and produce high-quality images in a few steps across three datasets: CIFAR-10 (32), AFHQ-Cat (256), and CelebA-HQ (256). The effectiveness of each component in our method is validated through ablation studies. Additionally, ASFM retains the beneficial capabilities of ODE-based generative models, such as zero-shot image editing. Our method offers a novel approach to attaining ODE-based generative models that produce straight flows based on FM, highlighting its potential applications in image generation tasks.

However, for more complex tasks like text-to-image generation, extending ASFM to large-scale datasets and enhancing the model's capability for controllable generation are necessary. In the future, we will focus on addressing these issues to further improve ASFM's performance.

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

## A    FLOW MATCHING WITH PAIRED DATA

**Proposition 1.** *Assuming that $p_t(\boldsymbol{x}) > 0$ for all $\boldsymbol{x} \in \mathbb{R}^n$ and $t \in [0,1]$, then, $\mathcal{L}_{CFM+}$ and $\mathcal{L}_{CFM}$ are equal, up to a constant independent of $\boldsymbol{\theta}$. Therefore, $\nabla_{\boldsymbol{\theta}} \mathcal{L}_{CFM+}(\boldsymbol{\theta}) = \nabla_{\boldsymbol{\theta}} \mathcal{L}_{FM}(\boldsymbol{\theta}) = \nabla_{\boldsymbol{\theta}} \mathcal{L}_{CFM}(\boldsymbol{\theta})$. And when flow (conditioned on $\boldsymbol{x}_0$ and $\boldsymbol{x}_1$) can be described as $\Gamma_t(\boldsymbol{x}) = (1-t)\boldsymbol{x}_0 + t\boldsymbol{x}_1$, $\mathcal{L}_{CFM+}$ is the same as 8.*

*Proof.* To prove this proposition, we first clarify the relationship between the conditional vector field and the marginal vector field.

Based on the properties of the conditional probability density function and continuity equation 3, we obtain:

$$
\begin{aligned}
\frac{d}{dt} p_t(\boldsymbol{x}) &= \iint \left( \frac{d}{dt} p_t(\boldsymbol{x}|\boldsymbol{x}_0, \boldsymbol{x}_1) \right) q(\boldsymbol{x}_0, \boldsymbol{x}_1) d\boldsymbol{x}_0 d\boldsymbol{x}_1 \\
&= -\iint \nabla \cdot \left( u_t(\boldsymbol{x}|\boldsymbol{x}_0, \boldsymbol{x}_1) p_t(\boldsymbol{x}|\boldsymbol{x}_0, \boldsymbol{x}_1) \right) q(\boldsymbol{x}_0, \boldsymbol{x}_1) d\boldsymbol{x}_0 d\boldsymbol{x}_1 \\
&= -\nabla \cdot \left( \iint u_t(\boldsymbol{x}|\boldsymbol{x}_0, \boldsymbol{x}_1) p_t(\boldsymbol{x}|\boldsymbol{x}_0, \boldsymbol{x}_1) q(\boldsymbol{x}_0, \boldsymbol{x}_1) d\boldsymbol{x}_0 d\boldsymbol{x}_1 \right).
\end{aligned}
\tag{15}
$$

According to continuity equation 3, we obtain:

$$
\frac{d}{dt} p_t(\boldsymbol{x}) = -\nabla \cdot (u_t(\boldsymbol{x}) p_t(\boldsymbol{x})).
\tag{16}
$$

Since the right-hand side of Equation 15 is equal to the right-hand side of Equation 16, we obtain:

$$
u_t(\boldsymbol{x}) = \frac{\iint u_t(\boldsymbol{x}|\boldsymbol{x}_0, \boldsymbol{x}_1) p_t(\boldsymbol{x}|\boldsymbol{x}_0, \boldsymbol{x}_1) q(\boldsymbol{x}_0, \boldsymbol{x}_1) d\boldsymbol{x}_0 d\boldsymbol{x}_1}{p_t(\boldsymbol{x})}.
\tag{17}
$$

In the following proof, we use similar setup as in Lipman et al. (2022): assuming that $q(\boldsymbol{x}_0, \boldsymbol{x}_1)$ and $p_t(\boldsymbol{x}|\boldsymbol{x}_0, \boldsymbol{x}_1)$ are decreasing to zero at a sufficient speed as $\|\boldsymbol{x}\| \to \infty$ to ensure existence of all integrals and to allow the changing of integration order (by Fubini's Theorem), and that $u_t, f_t, \nabla_{\boldsymbol{\theta}} f_t$ are bounded.

We will now prove that $\nabla_{\boldsymbol{\theta}} \mathcal{L}_{CFM+}$ and $\nabla_{\boldsymbol{\theta}} \mathcal{L}_{FM}$ are equal. Since $\nabla_{\boldsymbol{\theta}} \mathcal{L}_{FM}$ is known to be equal to $\nabla_{\boldsymbol{\theta}} \mathcal{L}_{CFM}$ (Lipman et al., 2022), we can conclude that $\nabla_{\boldsymbol{\theta}} \mathcal{L}_{CFM+} = \nabla_{\boldsymbol{\theta}} \mathcal{L}_{FM} = \nabla_{\boldsymbol{\theta}} \mathcal{L}_{CFM}$.

Expanding the 2-norms in $\mathcal{L}_{CFM+}$ and $\mathcal{L}_{FM}$, we obtain:

$$
\|f_t(\boldsymbol{x}) - u_t(\boldsymbol{x}|\boldsymbol{x}_0, \boldsymbol{x}_1)\|^2 = \|f_t(\boldsymbol{x})\|^2 - 2\langle f_t(\boldsymbol{x}), u_t(\boldsymbol{x}|\boldsymbol{x}_0, \boldsymbol{x}_1)\rangle + \|u_t(\boldsymbol{x}|\boldsymbol{x}_0, \boldsymbol{x}_1)\|^2,
\tag{18}
$$

$$
\|f_t(\boldsymbol{x}) - u_t(\boldsymbol{x})\|^2 = \|f_t(\boldsymbol{x})\|^2 - 2\langle f_t(\boldsymbol{x}), u_t(\boldsymbol{x})\rangle + \|u_t(\boldsymbol{x})\|^2.
\tag{19}
$$

The third terms on the right-hand side of Equation 18 and Equation 19 are independent of $\boldsymbol{\theta}$, so we will focus on the first two terms.

By deriving the first term on the right-hand side of both equations, we obtain:

$$
\begin{aligned}
\mathbb{E}_{t, p_t(\boldsymbol{x})} \|f_t(\boldsymbol{x})\|^2 &= \iint \|f_t(\boldsymbol{x})\|^2 p_t(\boldsymbol{x}) d\boldsymbol{x} dt \\
&= \iiiint \|f_t(\boldsymbol{x})\|^2 p_t(\boldsymbol{x}|\boldsymbol{x}_0, \boldsymbol{x}_1) q(\boldsymbol{x}_0, \boldsymbol{x}_1) d\boldsymbol{x} d\boldsymbol{x}_0 d\boldsymbol{x}_1 dt \\
&= \mathbb{E}_{t, q(\boldsymbol{x}_0, \boldsymbol{x}_1) p_t(\boldsymbol{x}|\boldsymbol{x}_0, \boldsymbol{x}_1)}.
\end{aligned}
\tag{20}
$$

By deriving the second term on the right-hand side of both equations, we obtain:

$$
\begin{aligned}
\mathbb{E}_{t, p_t(\boldsymbol{x})} \langle f_t(\boldsymbol{x}), u_t(\boldsymbol{x})\rangle &= \iint \langle f_t(\boldsymbol{x}), \frac{\iint u_t(\boldsymbol{x}|\boldsymbol{x}_0, \boldsymbol{x}_1) p_t(\boldsymbol{x}|\boldsymbol{x}_0, \boldsymbol{x}_1) q(\boldsymbol{x}_0, \boldsymbol{x}_1) d\boldsymbol{x}_0 d\boldsymbol{x}_1}{p_t(\boldsymbol{x})} \rangle p_t(\boldsymbol{x}) d\boldsymbol{x} dt \\
&= \iint \langle f_t(\boldsymbol{x}), \iint u_t(\boldsymbol{x}|\boldsymbol{x}_0, \boldsymbol{x}_1) p_t(\boldsymbol{x}|\boldsymbol{x}_0, \boldsymbol{x}_1) q(\boldsymbol{x}_0, \boldsymbol{x}_1) d\boldsymbol{x}_0 d\boldsymbol{x}_1 \rangle d\boldsymbol{x} dt \\
&= \iiiint \langle f_t(\boldsymbol{x}), u_t(\boldsymbol{x}|\boldsymbol{x}_0, \boldsymbol{x}_1)\rangle p_t(\boldsymbol{x}|\boldsymbol{x}_0, \boldsymbol{x}_1) q(\boldsymbol{x}_0, \boldsymbol{x}_1) d\boldsymbol{x}_0 d\boldsymbol{x}_1 d\boldsymbol{x} dt \\
&= \mathbb{E}_{t, q(\boldsymbol{x}_0, \boldsymbol{x}_1), p_t(\boldsymbol{x}|\boldsymbol{x}_0, \boldsymbol{x}_1)} \langle f_t(\boldsymbol{x}), u_t(\boldsymbol{x}|\boldsymbol{x}_0, \boldsymbol{x}_1)\rangle.
\end{aligned}
\tag{21}
$$

In this way, we prove that $\mathcal{L}_{CFM+}$ and $\mathcal{L}_{CFM}$ are equal up to a constant independent of $\boldsymbol{\theta}$. Therefore, $\nabla_{\boldsymbol{\theta}}\mathcal{L}_{CFM+}(\boldsymbol{\theta}) = \nabla_{\boldsymbol{\theta}}\mathcal{L}_{FM}(\boldsymbol{\theta}) = \nabla_{\boldsymbol{\theta}}\mathcal{L}_{CFM}(\boldsymbol{\theta})$.

When flow (conditioned on $\boldsymbol{x}_0$ and $\boldsymbol{x}_1$) can be described as $\Gamma_t(\boldsymbol{x}) = (1 - t)\boldsymbol{x}_0 + t\boldsymbol{x}_1$, we have that:

$$
\begin{aligned}
\frac{d}{dt}\Gamma_t(\boldsymbol{x}) &= \lim_{\Delta t \to 0} \frac{\left[\left((1 - (t + \Delta t))\right)\boldsymbol{x}_0 + (t + \Delta t)\boldsymbol{x}_1\right] - \left[(1 - t)\boldsymbol{x}_0 + t\boldsymbol{x}_1\right]}{\Delta t} \\
&= \boldsymbol{x}_1 - \boldsymbol{x}_0.
\end{aligned}
\tag{22}
$$

Therefore, when $\Gamma_t(\boldsymbol{x}) = (1 - t)\boldsymbol{x}_0 + t\boldsymbol{x}_1$, $u_t(\boldsymbol{x}|\boldsymbol{x}_0, \boldsymbol{x}_1) = \boldsymbol{x}_1 - \boldsymbol{x}_0$, and $\mathcal{L}_{CFM+}$ is the same as 8.

It is noteworthy that Equation 8 is consistent with the loss function used in Liu et al. (2022), indicating that the above proof establishes a connection between Flow Matching and Rectified Flow. This suggests that Rectified Flow can be viewed as a specific instance of Flow Matching with a designated conditional flow.

$\square$

# B  HYPERPARAMETERS FOR ASFM

We summarize the setting of hyperparameters in ASFM as shown in Table 4. On CIFAR-10, we trained two models: one without using an additional loss $\mathcal{L}_{Addition}$ during training, and the other using LPIPS loss as the additional loss $\mathcal{L}_{Addition}$.

Table 4: ASFM hyperparameters for different datasets.

| Dataset | Batch Size | Training Memory Size $k$ (batches) | Training Memory Update Frequency $freq$ | Maximum Training Steps $max\_step$ | ODE Solver | $\lambda_1$ | $\lambda_2$ | $\eta_1$ | $\eta_2$ |
|---|---|---|---|---|---|---|---|---|---|
| CIFAR-10 | 512 | 128 | 100 | 100,000 | RK45 | 0.01 | 0.0 | 2e-4 | 2e-4 |
| CIFAR-10 | 512 | 128 | 100 | 160,000 | RK45 | 0.01 | 1.0 | 2e-4 | 2e-4 |
| AFHQ-Cat | 64 | 128 | 100 | 120,000 | RK45 | 0.01 | 0.0 | 2e-4 | 2e-4 |
| CelebA-HQ | 64 | 128 | 100 | 120,000 | RK45 | 0.01 | 0.0 | 2e-4 | 2e-4 |

# C  ZEO-SHOT IMAGE EDITING

Despite incorporating Adversarial Training, ASFM fundamentally remains an ODE-based generative model, which enables it to perform certain zero-shot image editing tasks. Unlike GANs, which cannot find latent space representations for sampling points in the target distribution, ASFM, due to the reversible nature of ODEs, can identify corresponding sampling points in the initial distribution for each sampling point in the target distribution, thus serving as latent space representations. By manipulating these latent space representations, ASFM possesses the capability for zero-shot image editing. We will introduce the application of ASFM in image mixture in Section C.1.

## C.1  IMAGE MIXTURE

We selected an image of a yellow cat and an image of a white cat for mixing. Each image is split in half horizontally, with the upper half of the former combined with the lower half of the latter to create a mixture image. By using the mixture image as the initial value condition to solve the ODE in reverse (with NFE=300 to get the exact solution), we can obtain its latent space representation. Following Liu et al. (2022), we have two ways to manipulate the latent space representation: one is to multiply it directly by a coefficient $\alpha$, represented by $\boldsymbol{x}_0' = \alpha\boldsymbol{x}_0$; the other is to compute a weighted sum with a predetermined Gaussian noise $\epsilon$, represented by $\boldsymbol{x}_0' = \alpha\boldsymbol{x}_0 + \sqrt{1 - \alpha^2}\epsilon$. By using the processed latent space representation as the initial value condition to solve the ODE forward, the mixed image can be obtained. The results are illustrated in Figures 7 and 8.

When using the first operation method, a large $\alpha$ allows for a smooth blending of the two images. In the second method, since a predetermined Gaussian noise $\epsilon$ is incorporated, a smaller $\alpha$ results in the blended image being closer to the image corresponding to that noise.

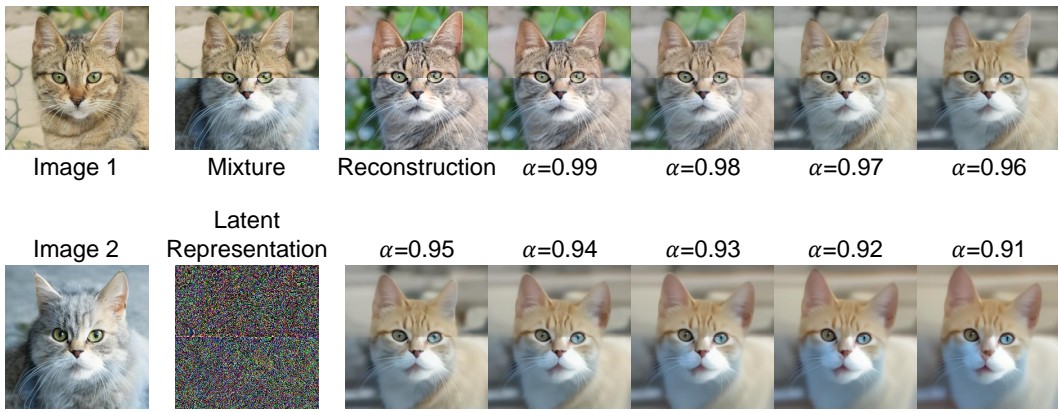

Figure 7: $\boldsymbol{x}_0' = \alpha \boldsymbol{x}_0$.

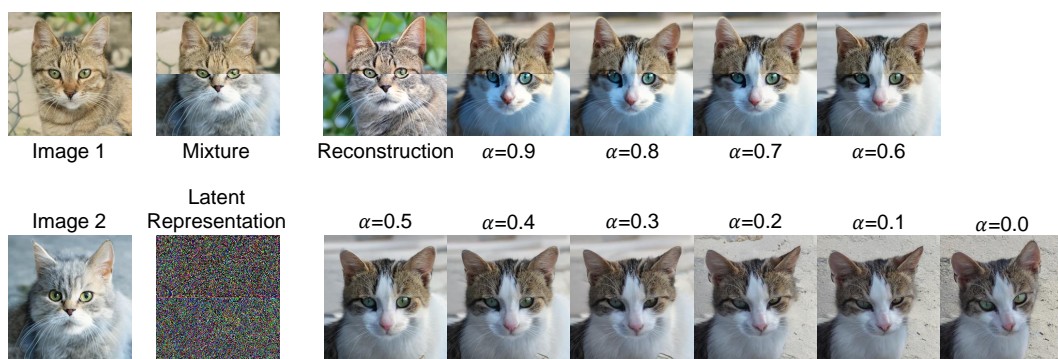

Figure 8: $\boldsymbol{x}_0' = \alpha \boldsymbol{x}_0 + \sqrt{1 - \alpha^2}\epsilon$.

## D INTERPOLATION

Due to the manifold hypothesis (Belkin & Niyogi, 2003; Roweis & Saul, 2000), image data is sparsely distributed in pixel space, making it challenging to achieve smooth and natural transitions between images through direct interpolation in pixel space. However, ASFM can connect the initial distribution (a Gaussian distribution) with the target distribution (the image data distribution) and consider the sampling points from the corresponding Gaussian distribution as latent space representations of the images. By interpolating in this latent space and then using ASFM to obtain the corresponding sampling points in the target distribution, smooth and natural transitions between images can be achieved.

As shown in Figure 9, by interpolating between the initial distribution sample point $\boldsymbol{x}_0$ and $\boldsymbol{x}_0'$ corresponding the target distribution sample point $\boldsymbol{x}_1$ and $\boldsymbol{x}_1'$, and then sampling through ASFM, we can achieve a high-quality and smooth transition process between $\boldsymbol{x}_1$ and $\boldsymbol{x}_1'$.

When interpolating between sample points $\boldsymbol{x}_0$ and $\boldsymbol{x}_0'$ of the initial distribution, we employed spherical linear interpolation, which ensures a smooth transition during the interpolation process. When applying spherical linear interpolation, first we compute the angle $\beta$ between two vectors $\boldsymbol{x}_0$ and $\boldsymbol{x}_0'$:

$$\beta = \cos^{-1}\left(\frac{\boldsymbol{x}_0 \cdot \boldsymbol{x}_0'}{\|\boldsymbol{x}_0\|\|\boldsymbol{x}_0'\|}\right). \tag{23}$$

The interpolation between $\boldsymbol{x}_0$ and $\boldsymbol{x}_0'$ is controlled by a parameter $\alpha$ in the range from 0 to 1. Then we computes the weights for each vector as follows:

$$w = \frac{\sin\big((1-\alpha)\beta\big)}{\sin(\beta)}$$
$$w' = \frac{\sin(\alpha\beta)}{\sin(\beta)}. \tag{24}$$

The interpolated result is shown as follows:

$$\boldsymbol{x}_0(\alpha) = w\boldsymbol{x}_0 + w'\boldsymbol{x}_0'. \tag{25}$$

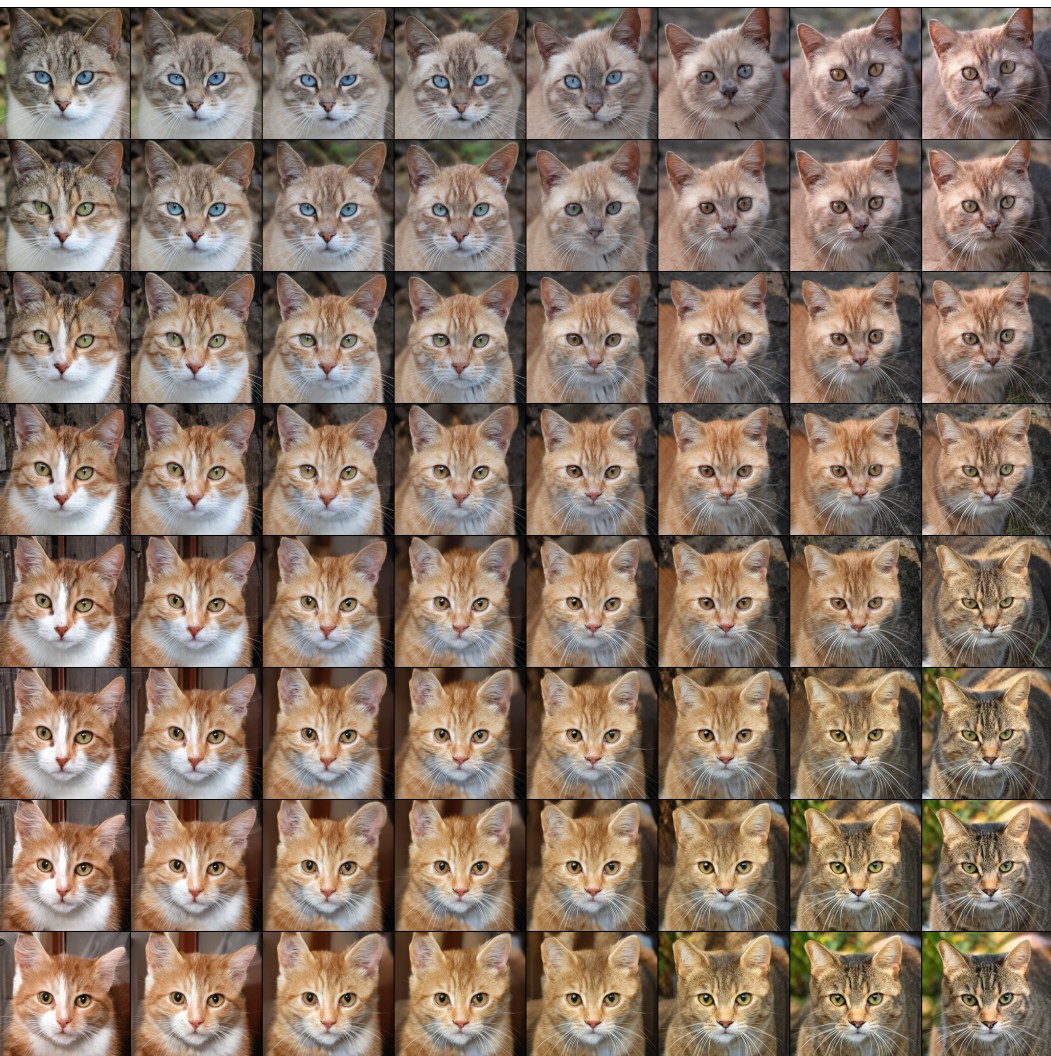

Figure 9: An illustration of interpolation among the images at the corners (sampling with NFE=6).

# E HYPERPARAMETER ABLATION STUDY

We conducted ablation studies on the hyperparameters in ASFM, including Training Memory Update Frequency $freq$ 10, $\lambda_1$ 11, and $\lambda_2$ 12, to validate the effectiveness of the parameters used in our experiments. We trained ASFM on the CIFAR-10 dataset for 50,000 steps and computed the corresponding FID by performing one-step image generation using the Euler solver.

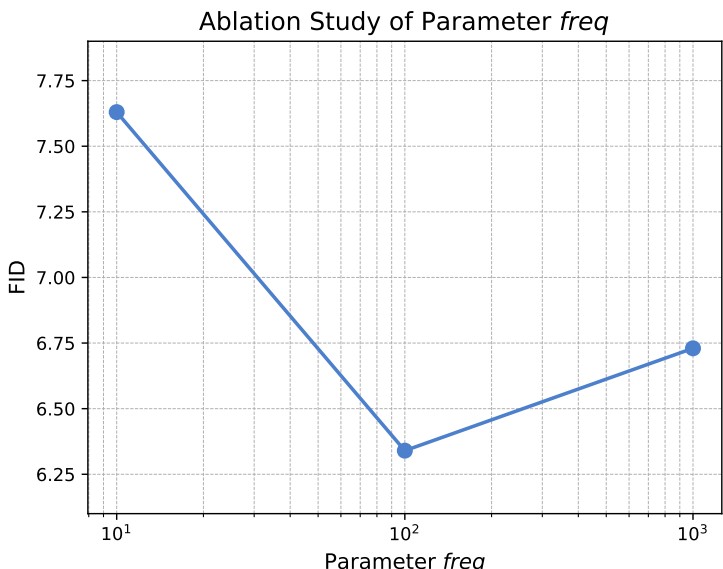

Figure 10: ablation study freq

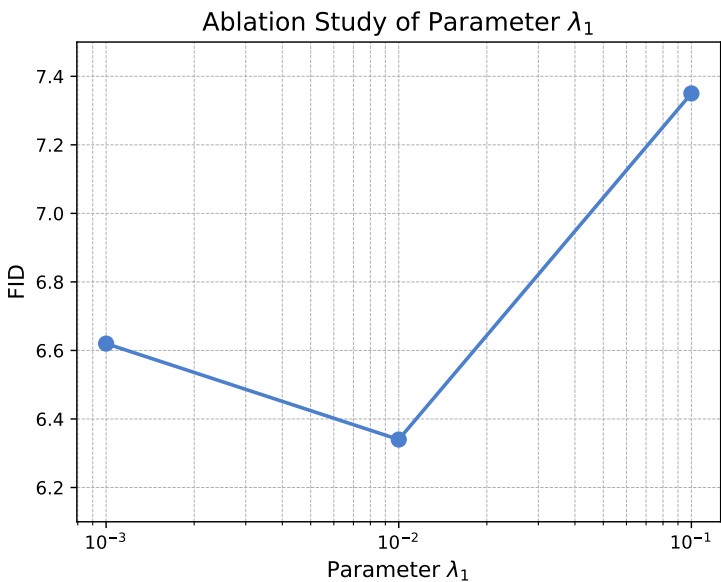

Figure 11: ablation study lambda 1

For the selection of the hyperparameter Training Memory Size $k$ (batches), we aim for the number of images in the Training Memory to slightly exceed the corresponding dataset size. For instance, on the CIFAR-10 dataset, the Training Memory contains $512 \times 128 = 65,536 \simeq 50,000 + 10,000$ (the size of CIFAR-10 train + CIFAR-10 test) images. Similarly, on the AFHQ-Cat dataset, the Training Memory contains $64 \times 128 = 8,192 \simeq 5,653$ (the size of AFHQ-Cat) images. However, due to the higher resolution and the large size of the CelebA-HQ dataset, which contains 30,000 images, generating excessive data can impose storage pressure. Therefore, we choose the Training Memory Size $k$ (batches) for CelebA-HQ to be consistent with that of AFHQ-Cat.

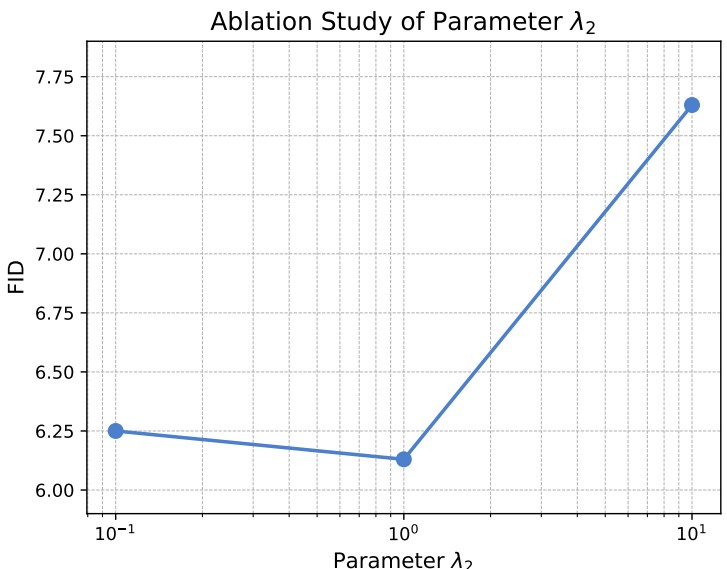

Figure 12: ablation study lambda 2

## F MORE GENERATED IMAGES

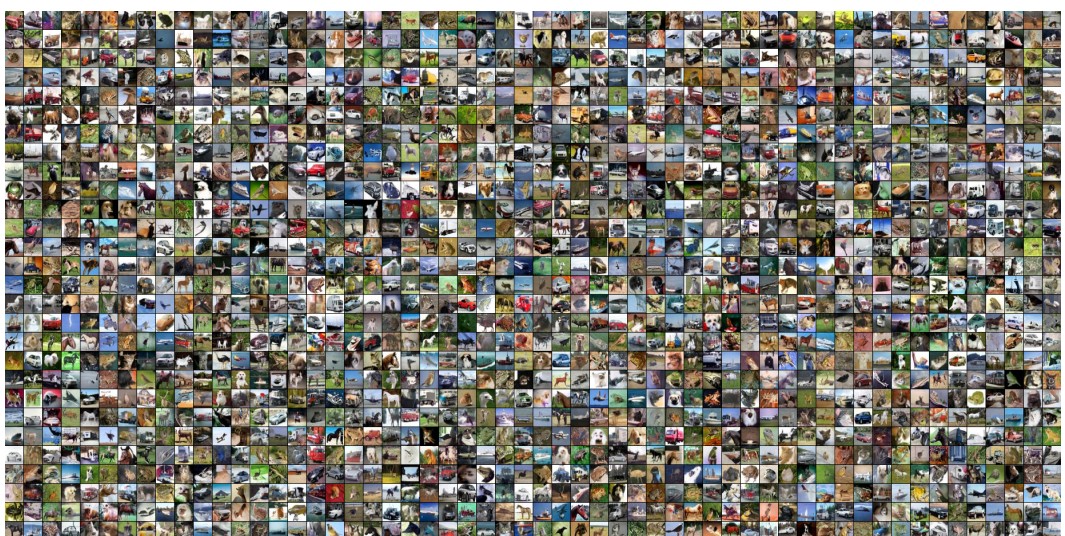

Figure 13: ASFM (1 NFE) on CIFAR-10.

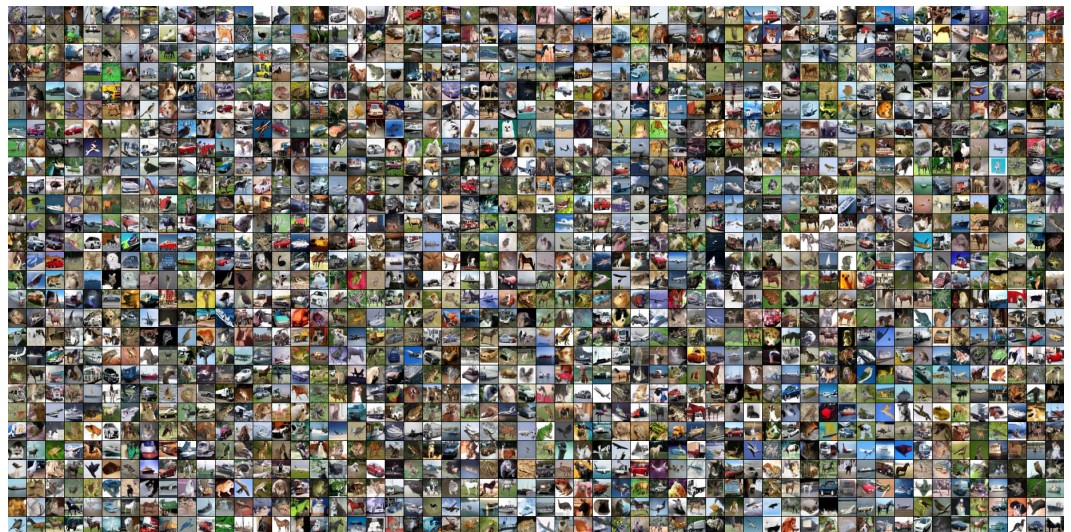

Figure 14: ASFM (NFE=2) on CIFAR-10.

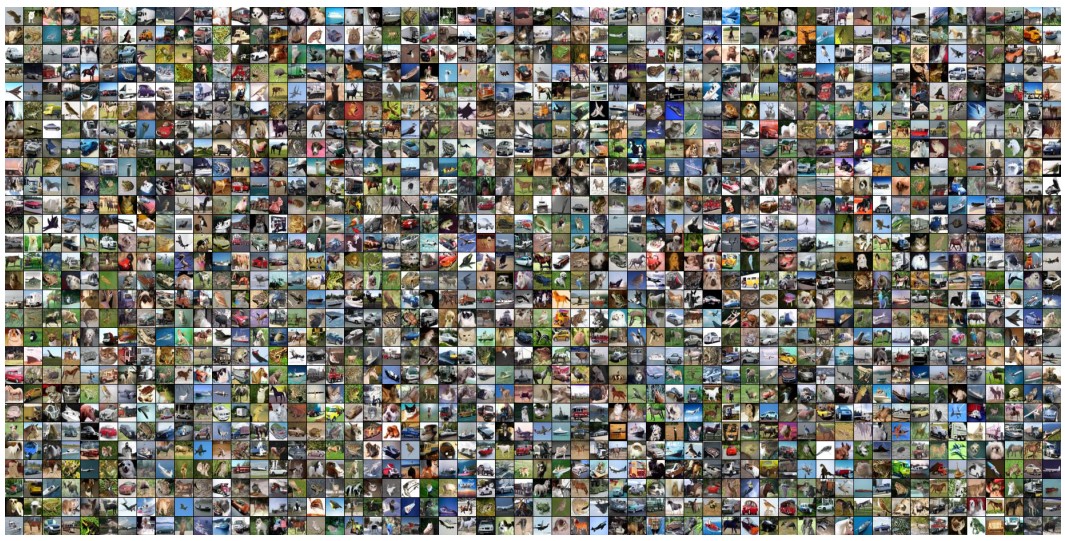

Figure 15: ASFM+lpips (1 NFE) on CIFAR-10.

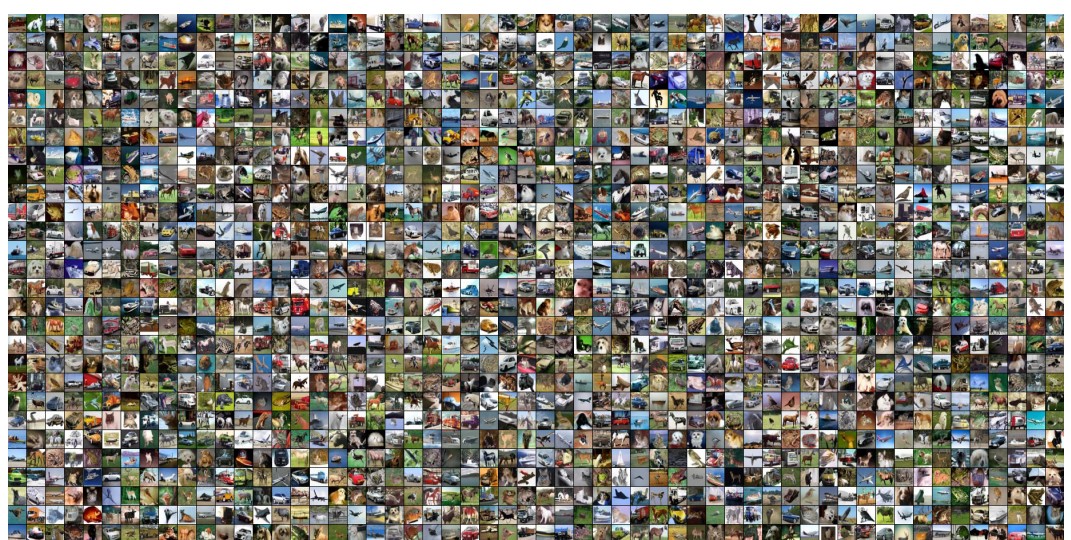

Figure 16: ASFM+lpips (NFE=2) on CIFAR-10.

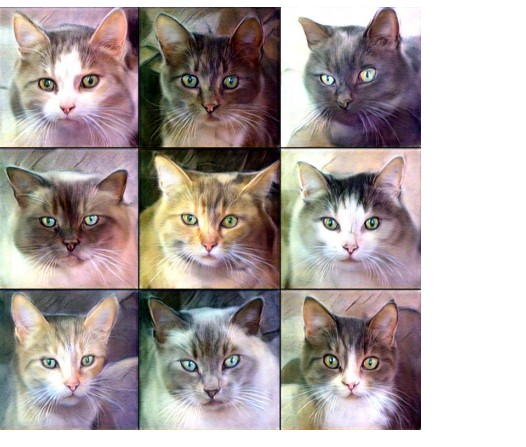
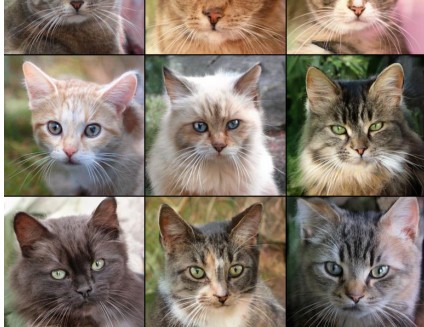

(a) ASFM (NFE=1) on AFHQ-Cat.                    (b) ASFM (NFE=2) on AFHQ-Cat.

Figure 17: Generated high-resolution images by ASFM, sampling with Euler Solver in 1 and 2 steps.

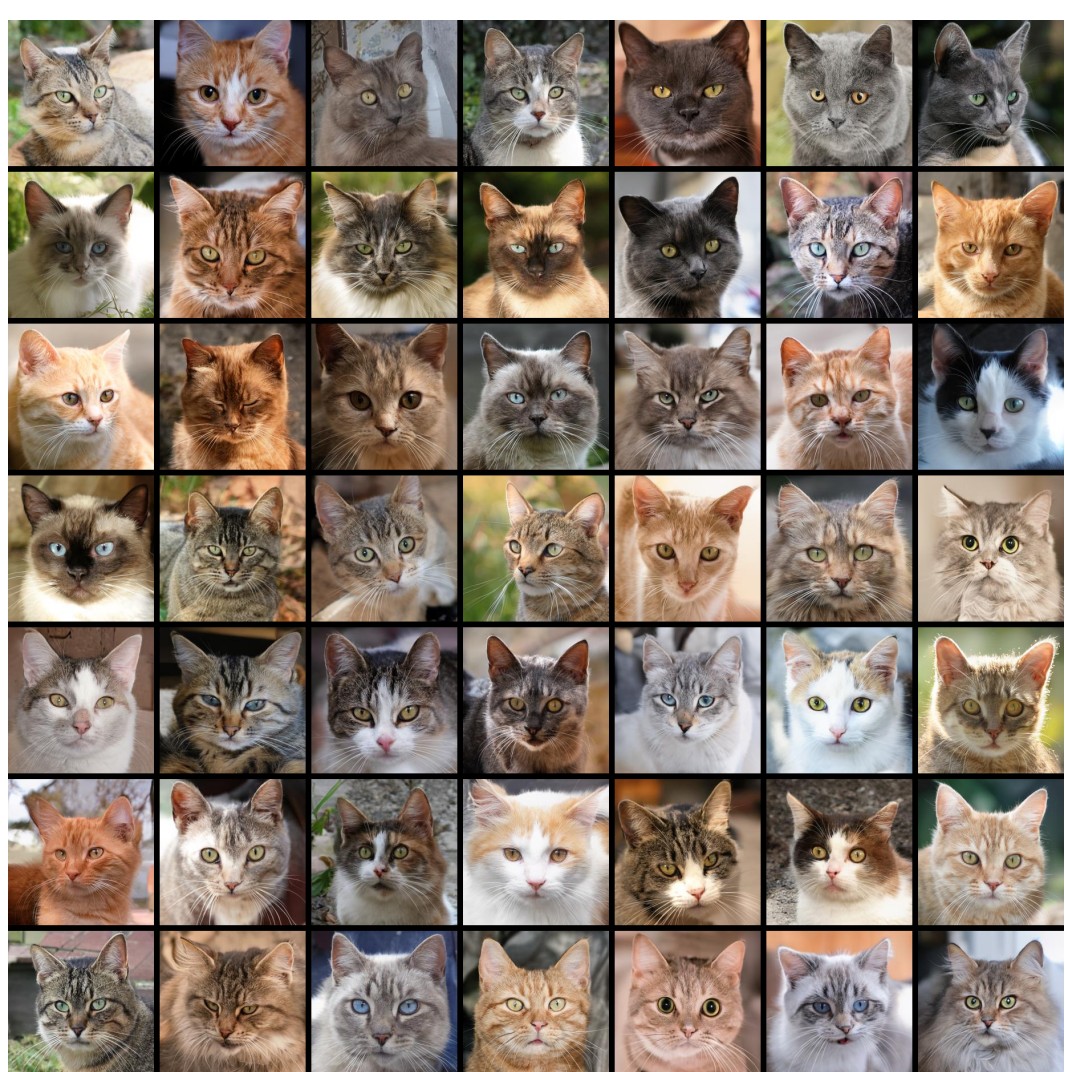

Figure 18: ASFM (NFE=6) on AFHQ.

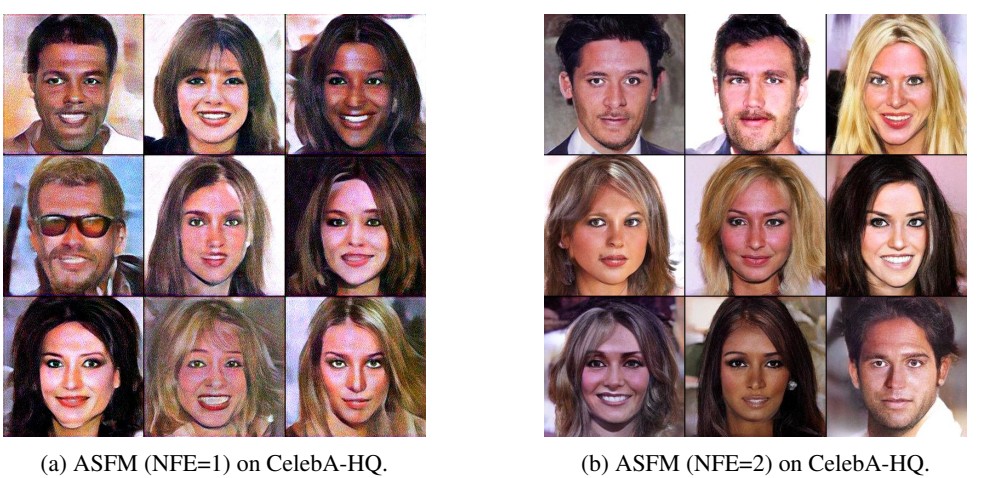

(a) ASFM (NFE=1) on CelebA-HQ.                    (b) ASFM (NFE=2) on CelebA-HQ.

Figure 19: Generated high-resolution images by ASFM, sampling with Euler Solver in 1 and 2 steps.

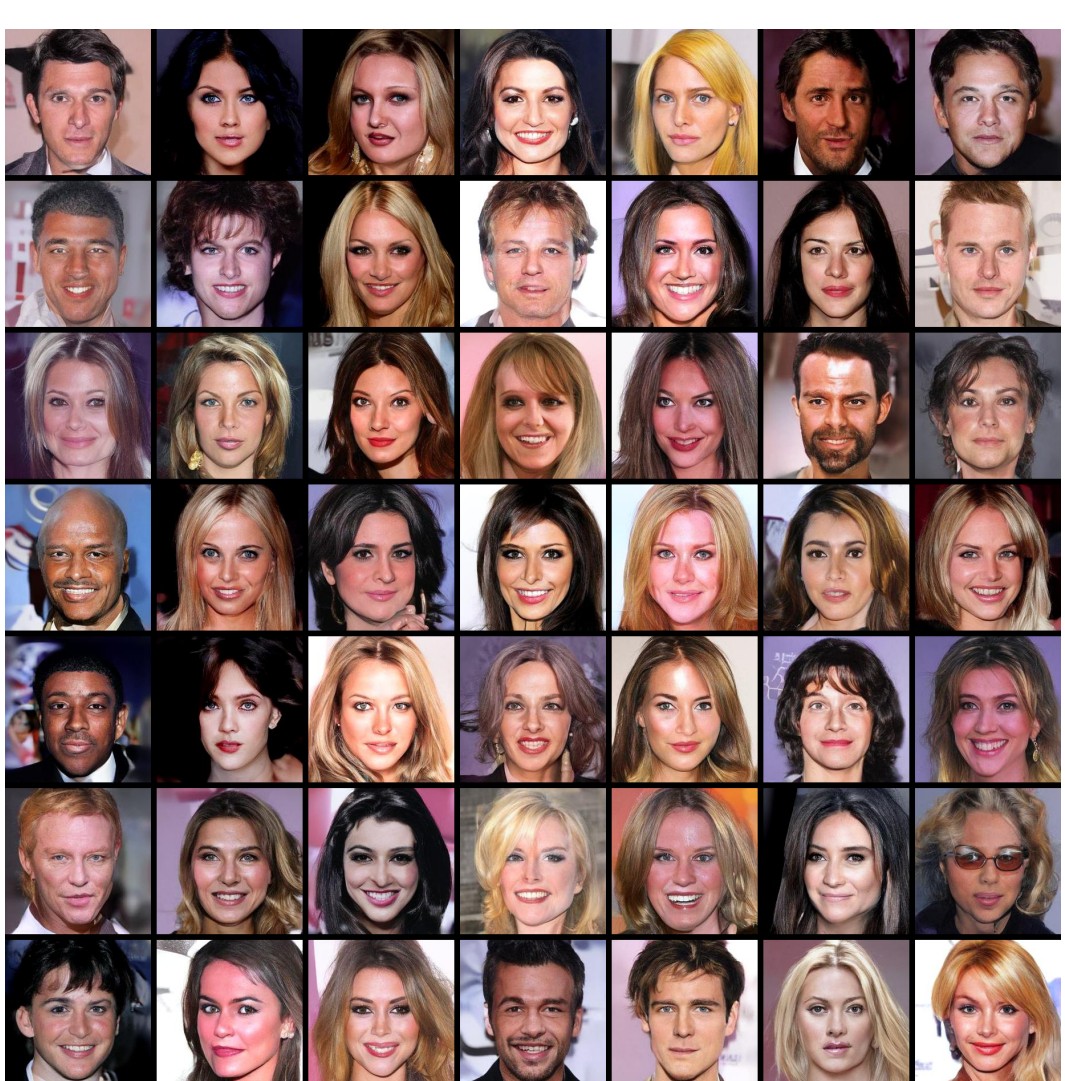

Figure 20: ASFM (NFE=6) on CelebA-HQ.

# G  BASELINE GENERATED IMAGES

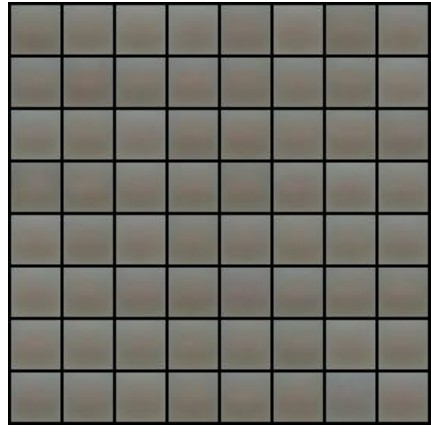

(a) 1-Rectified Flow (NFE=1) on CIFAR-10.

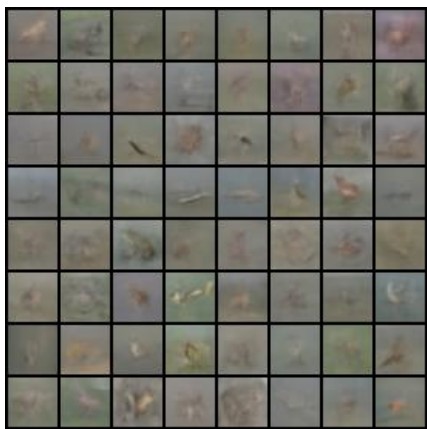

(b) 1-Rectified Flow (NFE=2) on CIFAR-10.

Figure 21: Generated images by 1-Rectified Flow, sampling with Euler Solver in 1 and 2 steps.

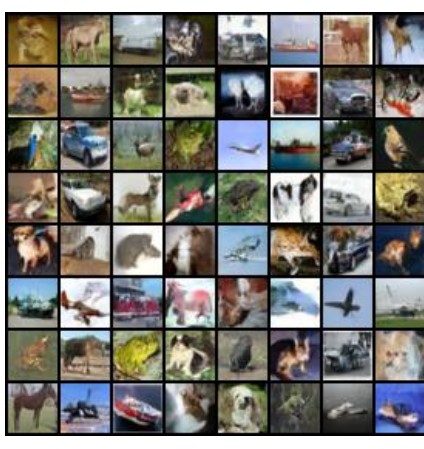

(a) 2-Rectified Flow (NFE=1) on CIFAR-10.

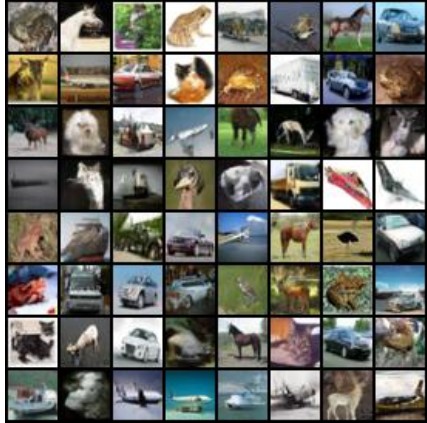

(b) 2-Rectified Flow (NFE=2) on CIFAR-10.

Figure 22: Generated images by 2-Rectified Flow, sampling with Euler Solver in 1 and 2 steps.

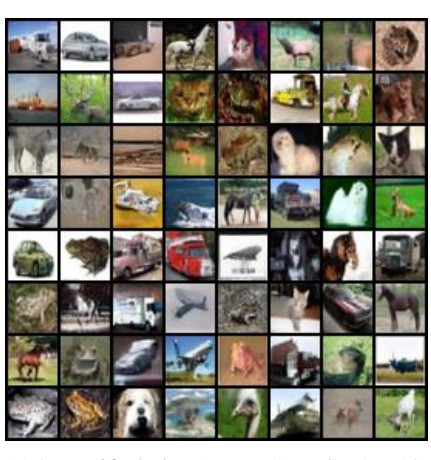
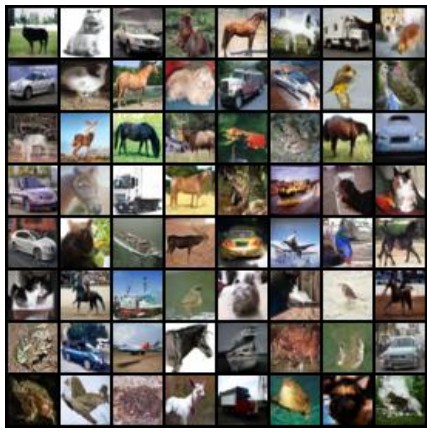

(a) 3-Rectified Flow (NFE=1) on CIFAR-10.    (b) 3-Rectified Flow (NFE=2) on CIFAR-10.

Figure 23: Generated images by 3-Rectified Flow, sampling with Euler Solver in 1 and 2 steps.

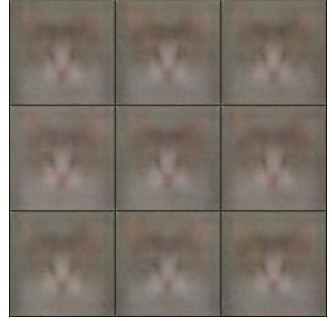
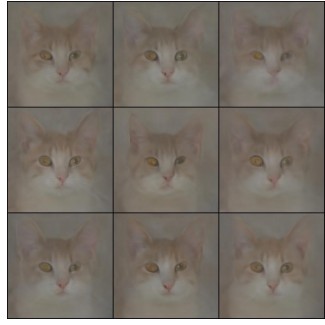
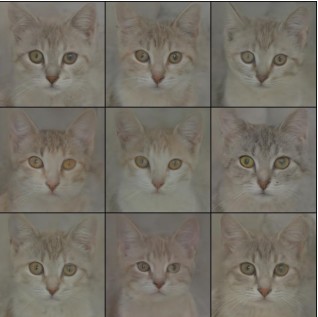

(a) 1-Rectified Flow (NFE=1) on AFHQ-Cat.    (b) 1-Rectified Flow (NFE=2) on AFHQ-Cat.    (c) 1-Rectified Flow (NFE=6) on AFHQ-Cat.

Figure 24: Generated images by 1-Rectified Flow, sampling with Euler Solver in 1, 2 and 6 steps.

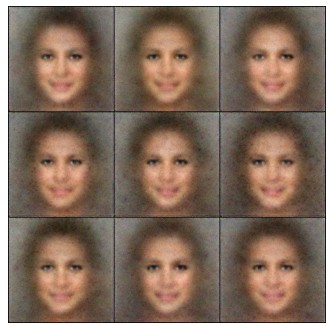
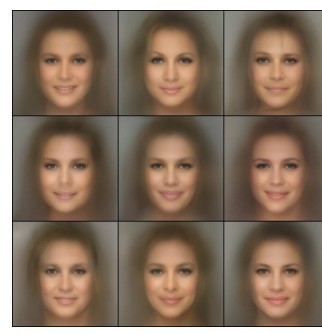
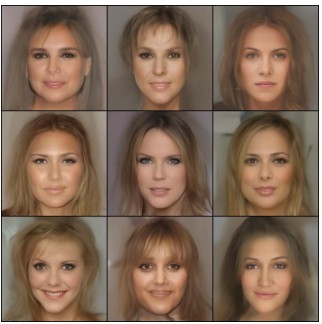

(a) 1-Rectified Flow (NFE=1) on CelebA-HQ.    (b) 1-Rectified Flow (NFE=2) on CelebA-HQ.    (c) 1-Rectified Flow (NFE=6) on CelebA-HQ.

Figure 25: Generated images by 1-Rectified Flow, sampling with Euler Solver in 1, 2 and 6 steps.

