# OpenReview forum: "Adversarial Self Flow Matching: Few-steps Image Generation with Straight Flows"
_ICLR.cc/2025/Conference — ICLR 2025 Conference Withdrawn Submission_

### Official Review · Reviewer_3iAH · 2024-11-01

**Soundness:** 2
**Presentation:** 2
**Contribution:** 2
**Rating:** 5
**Confidence:** 4

**Summary:**

This paper introduces ASFM as a novel Flow Matching model to straighten the trajectory of samples and generates high-quality images in a few steps. In contrast to previous approaches in the literature, such as the Reflow procedure and non-trivial pairing, ASFM learns flows using Adversarial Training and Online Self Training to straighten flows and generates high-quality images in a few steps. Experiments are conducted on CIFAR-10, CelebA-HQ (256) and AFHQ-Cat (256) dataset.

**Strengths:**

The paper is clear and easy to follow.

The core idea of ASFM that introduces adversarial training after online self training is interesting.

**Weaknesses:**

ASFM can achieve good FID scores on CIFAR-10, but the generated images in Fig.4 are not good enough. More generated images from other baselines are needed to support the experiment.

While the authors assert that their research exhibits the one-step generation capability in the abstract, the existing experimental results do not sufficiently validate this claim: No high- resolution sample with fewer-steps (NFE<6) is displayed.

The authors also assert that ASFM is a straight flow, but the visual results do not sufficiently validate this claim. I encourage the authors to show the flow is straight over the entire time span, i.e., generated samples with different sampling steps and the same initial noise.

**Questions:**

The clarity of Fig.2 is insufficient. The transparent red dashed line is some what confusing. It would benefit from additional labels or a more detailed legend to help readers better understand its significance.

The image mixture described in Appendix.C.1 is not clear and the results in Fig.7-8 are not convincing , making it difficult for me to understand the purpose. Providing more detailed explanations or context would greatly improve comprehension.

---

> ### Author Response · Authors · 2024-11-22
>
> We appreciate the reviewer's comments, and we provide clarifications on these issues below.
>
> **Q1: More generated images from other baselines.**
>
> **A1:** We provide 1-Rectified Flow, 2-Rectified Flow and 3-Rectified Flow generated images with NFE=1and NFE=2 on CIFAR-10 for comparison in Appendix G.
>
> **Q2: High-resolution images generated by ASFM with fewer steps.**
>
> **A2:** We provide ASFM generated images on CelebA-HQ and AFHQ-Cat with NFE=1 and NFE=2 in Appendix F and we also provide 1-Rectified Flow generated images on CelebA-HQ and AFHQ-Cat with NFE=1, NFE=2 and NFE=6 in Appendix G.
>
> **Q3: ASFM can generated straight flows.**
>
> **A3:** ASFM's ability to generate straight flows is directly evidenced by its capacity to produce high-quality generated images with very few sampling steps. This is because only sufficiently straight flows can yield accurate solutions with minimal steps when solving the ODE numerically. Additionally, ASFM's performance on the CIFAR-10 dataset, where it achieves results very close to those obtained with 113 steps using only two steps, further demonstrates its capability to generate straight flows, as shown in the following table.
>
> | solver | NFE  | FID  | IS   |
> | ------ | ---- | ---- | ---- |
> | euler  | 2    | 4.82 | 8.93 |
> | rk45   | 113  | 4.80 | 8.96 |
>
> **Q4: The clarity of Figure 2.**
>
> **A4:** We have clarified the meaning of the transparent red dashed line in Figure 2, indicating that it represents the vector field trained with paired data $(x_0, x_1)$ and $(x_0', x_1')$ which have not been repaired. The Self Online Training process illustrated in Figure 2 can be understood as follows: 1) Initially, during the training of the model, if there are intersections in the connections between paired data$(x_0, x_1)$ and $(x_0', x_1')$, the learned vector field $f_t$ (the transparent red dashed line) will approach the intersection point. 2) Due to the non-crossing nature of flow fields, the learned vector field will generate data that repairs the previously paired data, resulting in repaired data $(x_0, x_1')$ and $(x_0', x_1)$ . 3)The connections of the repaired data no longer have intersections, and the newly learned vector field $f_t'$ (the opaque red dashed line) will become straighter.
>
> **Q5: The clarity of mixture experiment.**
>
> **A5:** The mixture experiment is designed to demonstrate ASFM's zero-shot image editing ability, meaning its capability to perform image editing without additional training. In the mixture experiment, we selected two images of cats. Simply stitching these two images together results in an unnatural outcome. However, using ASFM, we can obtain the latent space representations of the two images (by solving the ODE backward to find their corresponding points in the initial distribution). By stitching these two latent space representations together, adding some noise, and passing them through ASFM, we can generate a more natural mixed image. This mixed image retains features from both original cat images, such as a yellow upper half and a white lower half.

---

> > ### Comment · Reviewer_3iAH · 2024-11-27
> >
> > Thank you for your response. Some of my concerns have been addressed, so I have adjusted my rating accordingly.

---

### Official Review · Reviewer_oeao · 2024-11-02

**Soundness:** 2
**Presentation:** 2
**Contribution:** 2
**Rating:** 3
**Confidence:** 5

**Summary:**

The paper introduce adversarial training for flow matching framework that better aligns the one-step generated solution and real data. The method is shown to reduce the cumulative errors of solution trajectories of flow rectification.

**Strengths:**

- Leverage adversarial objective to align one-step prediction with real data distribution
- The idea is simple and intuitive while showcasing their strong performance on different datasets.

**Weaknesses:**

- The method has limited novelty as the usage of adversarial training in one-step generation is introduced in several works like ASM [1], CTM [2], ADD [3]. The method is solely the combination between flow-rectification and adversarial training.
- The effectiveness of the methods appears limited to low-resolution data like CIFAR-10, as shown in Table 2, where the FID scores for 1 and 2 NFE remain quite high. Additionally, the table omits the results for 2-rectified-flow and 3-rectified-flow, which is shown to achieve better FID scores under few-steps sampling (similar to the observation in Table 1). The authors should ensure fairness in their reporting so please include the results of 2-rectified-flow and its distillation in the table.
- Need to update table 2 with some few-steps works like CTM & CD as well.
- Table 1: the authors should report results of RectifiedFlow in combination with their distilled versions for 1-NFE generation as they give best FID scores of 6.18 vs 378 for 1-rectified-flow, 4.85 vs 12.21 for 2-rectified-flow, and 5.21 vs 8.15 for 3-rectified-low. Again, the authors should be integrity in their reports.
- The adversarial training is supposed to improve the gap between data distribution and one-step prediction. However, for high-resolution dataset as in table 2, the method still need more than one-step to produce better FID score under 10.0.
- Meanwhile, the authors should report the FID and NFE of the original flow matching model that they build upon for better exposition. In L332, it showed that the method uses the pretrained 1-rectified-flow. However, the 1-rectified-flow is shown to have straighter solution strajectories than the original flow model. It is reasonable for the authors to test their method starting with the original flow model.
- The method requires online-sampling of new samples to put in the queue. Does it add cost to the training progress in terms of speed and computation? Besides, the method requires two-separate model forward passes for flow matching loss and adversarial loss as shown in Algorithm 1. What if both objectives using the same random t1, how is the model performance?
- Is the size of training memory associating with model performance? The paper is only test with a size of 128 batches.
- The ablation of number of generated samples in Table 3 should include the case where 2-rectified-flow-online use the same amout of data size to compare with ASFM.
- Ablation of $\lambda_1, \lambda_2$ in Eq. 13 is missing.
- Zeroshot editing task should be moved to the main manuscript.

[1] Jolicoeur-Martineau, Alexia, et al. "Adversarial score matching and improved sampling for image generation." arXiv preprint arXiv:2009.05475 (2020).

[2] Kim, Dongjun, et al. "Consistency trajectory models: Learning probability flow ode trajectory of diffusion." in ICLR 2024.

[3] Sauer, Axel, et al. "Adversarial diffusion distillation." in ECCV 2025.

**Questions:**

Please address my concern above.

---

> ### Author Response · Authors · 2024-11-22
>
> We sincerely appreciate the reviewer for the constructive and insightful comments, and we will provide detailed responses to these issues below.
>
> **Q1: Adversarial training in one-step generation has already been used in some works.**
>
> **A1:** First, we would like to emphasize that our goal is **not to achieve one-step generation** but rather to utilize adversarial training to narrow the gap between the generated data distribution and the real data distribution. By combining this with Online Self Training, we aim to develop ODE-based generative models capable of **generating straight flows**, as indicated in the paper's title. Second, the references provided by the reviewer are fundamentally different from ASFM. CTM is based on consistency models; ADD is a distillation-based method; and ASM introduces adversarial training into score matching. The methods proposed in these works differ significantly from Flow Matching. To the best of our knowledge, ASFM is the first to apply adversarial training to Flow Matching in image generation, achieving highly competitive results among various Flow Matching-based methods.
>
> **Q2: The effectiveness of ASFM.**
>
> **A2:** Under the same number of NFEs, ASFM achieves significantly better generation performance on CelebA-HQ and AFHQ-Cat compared to other Flow Matching-based methods. As shown in Table 2, with NFE=6, ASFM reduces the FID on the AFHQ-Cat dataset by **at least 34%** compared to other methods (ASFM vs. BOSS) and reduces the FID on the CelebA-HQ dataset by **at least 56%** (ASFM vs. BOSS). This directly demonstrates the effectiveness of ASFM on high-resolution datasets.
>
> ASFM's one-step generation results on high-resolution datasets outperform the results of 1-Rectified Flow with six steps, and ASFM's two-step generation results are either superior to or comparable with Consistency-FM's six-step results. This highlights the superior image quality generated by ASFM under extremely few sampling steps. Rectified Flow did not conduct experiments with 2-rectified flow or 3-rectified flow on high-resolution datasets such as CelebA-HQ (256x256) and AFHQ-Cat (256x256). Thus, we are unclear about the reviewer's statement that Rectified Flow has been "shown to achieve better FID scores under few-steps sampling."
>
> On the CIFAR-10 dataset, with NFE=2, ASFM demonstrates significantly better generation performance than 2-Rectified Flow and 3-Rectified Flow, as shown in Table 1. Moreover, on CIFAR-10, the distilled results of 1-Rectified Flow (6.18), 2-Rectified Flow (4.85), and 3-Rectified Flow (5.21) are very close to ASFM's generation result with NFE=1 (5.07), although we believe comparing ASFM with **distillation-based** methods is not entirely fair.
>
> **Q3: Update Table 2 with some works unrelated to Flow Matching.**
>
> **A3:** ASFM is consistently based on Flow Matching, offering advantages not found in other methods such as CTM. For example, ASFM is theoretically grounded in ODE-based generative models. Due to the reversible nature of ODEs, ASFM can identify the latent space representation of any image within the initial distribution, facilitating tasks like image interpolation and zero-shot image editing, as demonstrated in Appendices C and D. In contrast, methods based on consistency models, such as CTM, lack an explicit ODE solver, complicating the inversion process. Given the fundamental theoretical differences between these methods and ASFM, comparing ASFM with them holds significant value.
>
> **Q4: Add distilled rectified flow results in Table1.**
>
> **A4:** Same as our response A2, the distilled results of 1-Rectified Flow (6.18), 2-Rectified Flow (4.85), and 3-Rectified Flow (5.21) are very close to ASFM's generation result with NFE=1 (5.07), although we believe comparing ASFM with **distillation-based** methods is not entirely fair.

---

> ### Author Response · Authors · 2024-11-22
>
> **Q5: On high-resolution datasets, the FID of images generated by ASFM with NFE=1 is greater than 10.0.**
>
> **A5:** We are not sure about the basis for the standard that the image quality with NFE=1 should achieve an FID below 10.0. If possible, we kindly ask the reviewer to provide the source for this standard. Furthermore, we believe that, as shown in Table 2, ASFM has already demonstrated outstanding results compared to other methods.
>
> **Q6: ASFM does not use "original flow model".**
>
> **A6:** We believe that the naming convention of Rectified Flow might have caused some confusion for the reviewer. The "1-rectified flow" could be what the reviewer referred to as the "origin flow model," as 1-rectified flow does not use the Rectify operation. Only when $n \geq 2$ does the n-rectified flow employ the Rectify operation.
>
> **Q7: Computation of online sampling and using same t in Algorithm 1.**
>
> **A7:** Compared to the Rectify operation proposed in Rectified Flow, ASFM's online sampling incurs lower computational overhead. For instance, as mentioned in the official implementation on Rectified Flow's GitHub repository, they recommend generating at least $1,000,000$ pairs of data for reflow on the CIFAR-10 dataset. In contrast, ASFM requires at most $(1600+127)×512=884,224$ pairs of data on CIFAR-10.
>
> Regarding the use of a random $t$ for the two interpolations in Algorithm 1, we believe that as training progresses, using the same $t$ or different $t$ values will sufficiently sample $t$, so the impact on training should be minimal. Nevertheless, this is an approach worth exploring. However, due to time constraints, we leave this for future work.
>
> **Q8: The size of Training Memory.**
>
> **A8:** For the selection of the hyperparameter Training Memory Size $k$ (batches), we aim for the number of images in the Training Memory to slightly exceed the corresponding dataset size. For instance, on the CIFAR-10 dataset, the Training Memory contains $512 \times 128 = 65,536 \simeq 50,000 + 10,000$ (the size of CIFAR-10 train + CIFAR-10 test) images. Similarly, on the AFHQ-Cat dataset, the Training Memory contains $64 \times 128 = 8,192 \simeq 5,653$ (the size of AFHQ-Cat) images. However, due to the higher resolution and the large size of the CelebA-HQ dataset, which contains 30,000 images, generating excessive data can impose storage pressure. Therefore, we choose the Training Memory Size $k$ (batches) for CelebA-HQ to be consistent with that of AFHQ-Cat.
>
> **Q9: Ablation study of $\lambda_1$ and $\lambda_2$.**
>
> **A9:** Thank the reviewer for this comment. We have added the relevant ablation experiments in Appendix E.

---

> > ### Comment · Reviewer_oeao · 2024-11-24
> >
> > > Q5: On high-resolution datasets, the FID of images generated by ASFM with NFE=1 is greater than 10.0.
> >
> > I believe the generation quality is not particularly impressive. When comparing the one-step and two-step results on CIFAR-10 (5.80 vs 4.82), the gap between these settings is quite small. However, for high-resolution data like Celeb, the gap is much larger (79.0 vs 22.2), which raises concerns that the method may be limited to low-resolution data. The same concern is also raised by reviwer `3iAH`.
> >
> >
> > Overall, I still keep my score at 3 as the paper requires significant revisions with more comprehensive experiments. Additionally, the method needs substantial improvements to meet the standards of a high-quality publication, given its limitations in novelty. Therefore, I believe the paper is not ready for this conference and lean toward rejection.

---

> > > ### Author Response · Authors · 2024-12-01
> > >
> > > We would like to express our sincere gratitude to the reviewer for the time and effort they have dedicated to reviewing our work.
> > >
> > > In response to the reviewer's comments, we have clarified some issues in ASFM and added experiments. The reviewer pointed out that ASFM should be initialized using the original flow model, which stems from a misunderstanding due to the naming of Rectified Flow. The initialization model we used is indeed the "original flow model," and we have provided an explanation for this issue.
> > >
> > > Additionally, the reviewer expressed concerns about the computational cost of using online training in ASFM. However, after careful calculation, we found that the amount of data required for ASFM's online training is smaller than the data required for reflow in Rectified Flow, demonstrating that the computational cost of ASFM's online training is reasonable.
> > >
> > > The reviewer also raised questions about the size of the Training Memory in ASFM. We have addressed this by clarifying that the size of the Training Memory primarily depends on the dataset size and storage constraints.
> > >
> > > Furthermore, the reviewer requested that we conduct ablation experiments on some hyperparameters in ASFM. We have completed these experiments and included the results in Appendix E.
> > >
> > > In addition to the issues already addressed above, the reviewer raised concerns about the effectiveness of ASFM, primarily in the following areas.
> > >
> > > First, the reviewer requested that we compare ASFM with other non-flow matching-based methods that utilize adversarial training for one-step image generation. It is important to note that, while works like CTM, as mentioned by the reviewer, have achieved impressive results in one-step generation, ASFM has features that these methods do not possess. Specifically, the generation process of ASFM is based on an ODE (Ordinary Differential Equation), and the reversible nature of the ODE allows for bidirectional transformations between the initial distribution sampling points and the target distribution sampling points. When solving the ODE in reverse, ASFM provides a corresponding sample point from the initial distribution for each sample point in the target distribution, which can serve as a latent space representation. This representation greatly facilitates zero-shot image editing, as detailed in Appendix C and D. This is a unique advantage that methods related to distillation, such as CTM, do not possess. Furthermore, it is important to clarify that while flow matching and score-based methods are closely related, they still differ significantly. A diffusion model can often be considered a special case of flow matching, as discussed in [1], Section 3.5, [2], Section 4.2, and [3], Section 2.2. The reviewer's statement that "flow matching and score-based methods have been proven equivalent" is not always accurate, though there are some special cases of flow matching are equivalent to score-based methods, as shown in [4], Section 1: "This holds for, for example, the v-prediction loss and flow matching with the optimal transport path." Hence, Applying adversarial training in flow matching is an area worth exploring. Moreover, ASFM has been designed with unique modules and methods, such as Training Memory and Online Self Training, to successfully incorporate adversarial training into flow matching, and these designs represent technical innovations.
> > >
> > > Second, the reviewer’s comments focused on the effectiveness of ASFM on high-resolution datasets. They requested that we compare ASFM with n-rectified flow (n > 1) on high-resolution datasets. However, Rectified Flow has not conducted related experiments, and due to time constraints, we were unable to perform these experiments during the rebuttal period but we will add some experiments to the final paper. Nevertheless, our method has already demonstrated superior properties in comparison with other methods, as shown in Table 2.
> > >
> > > [1] Liu, X., Gong, C., & Liu, Q. (2022). Flow straight and fast: Learning to generate and transfer data with rectified flow. *arXiv preprint arXiv:2209.03003*.
> > >
> > > [2] Lipman, Y., Chen, R. T., Ben-Hamu, H., Nickel, M., & Le, M. (2022). Flow matching for generative modeling. *arXiv preprint arXiv:2210.02747*.
> > >
> > > [3] Albergo, M. S., & Vanden-Eijnden, E. (2022). Building normalizing flows with stochastic interpolants. *arXiv preprint arXiv:2209.15571*.
> > >
> > > [4] Kingma, D., & Gao, R. (2024). Understanding diffusion objectives as the elbo with simple data augmentation. *Advances in Neural Information Processing Systems*, *36*.

---

> ### Comment · Reviewer_oeao · 2024-11-24
>
> Thank you for addressing my questions. However, I am not fully satisfied with the authors’ rebuttal.
>
> > Q1: Adversarial training in one-step generation has already been used in some works.
>
> The explanation is not convincing to me. While the goal of the proposed method may extend beyond one-step generation, its primary focus remains on minimizing the gap between real and generated data in fewer-step settings. Therefore, comparisons with other one-step and few-step generation methods (e.g. CTM and CD) are valid and important, not necessarily limited to flow-matching methods. So, the argument "Q3: Update Table 2 with some works unrelated to Flow Matching." does not make sense at all. Furthermore, the authors overstate their contribution by claiming to be the first to use adversarial training for flow matching. It is important to note that flow matching and score-based methods have been proven equivalent in [1], which shows minimal differences between the proposed method and the ASM paper.
>
> > Q2: The effectiveness of ASFM.
>
> For CIFAR10, the argument "Moreover, on CIFAR-10, the distilled results of 1-Rectified Flow (6.18), 2-Rectified Flow (4.85), and 3-Rectified Flow (5.21) are very close to ASFM's generation result with NFE=1 (5.07), although we believe comparing ASFM with distillation-based methods is not entirely fair." is not valid. Since, your method can be considered as a different type of flow straightening, the proposed method still benefits from a further distillation stage. So, the comparisons with distillation-based models are strongly necessary.
>
> For high-res data, the authors’ arguments for neglecting the results of RectifiedFlow on Celeb and AFHQ-Cat are not valid. As RectifiedFlow is a key baseline for the proposed method, the authors should provide these missing results (even if they were not originally included in the baseline) to ensure a fair comparison.
>
> Regarding the statement, “shown to achieve better FID scores under few-steps sampling,” my observation on CIFAR-10 indicates that multi-step rectification yields better model performance under the same NFE settings. Therefore, I believe this observation also holds for high-resolution datasets like Celeb and AFHQ-Cat, as supported by [2, 3].
>
> > Q4: Add distilled rectified flow results in Table1.
>
> Once again, this excuse is not valid. Please refer to my concerns above.
>
> [1] Understanding Diffusion Objectives as the ELBO with Simple Data Augmentation, https://arxiv.org/abs/2303.00848
> [2] Scaling Rectified Flow Transformers for High-Resolution Image Synthesis, https://arxiv.org/abs/2403.03206
> [3] InstaFlow: One Step is Enough for High-Quality Diffusion-Based Text-to-Image Generation, https://arxiv.org/abs/2309.06380v2

---

### Official Review · Reviewer_7v97 · 2024-11-03

**Soundness:** 3
**Presentation:** 3
**Contribution:** 2
**Rating:** 5
**Confidence:** 3

**Summary:**

This paper improves flow matching-based models by integrating two training schemes: (1) Online Self-training and (2) Adversarial Training, which are combined and called Adversarial Self Flow Matching (ASFM). (1) is an upgraded version of the original reflow (from Rectified Flow paper) by converting it into an online learning strategy, where the synthetic data is generated on the fly, instead of generated after each rectified flow process. (2) is a way to better align the synthetic data distribution with the real data distribution by using GAN loss. The author validates ASFM's effectiveness through multiple datasets such as CIFAR-10, CelebA-HQ, and AFHQ-Cat showing competitive scores using few denoising steps (NFE=1 to 6). Furthermore, the authors show downstream task applications such as image editing are also applicable upon the trained models.

**Strengths:**

Presentation:
The paper is well-structured and provides clear explanations of both the conceptual foundation and technical implementations of ASFM. Key technical components, such as the loss functions and training algorithm, are detailed sufficiently to allow reproducibility and insight into the model’s design choices.

The paper introduces a new method that integrates GAN loss into flow-matching, along with new strategy to implement reflow. The method is also experimented quite extensively through multiple datasets and the trained models have good quality only using a few denoising steps.

**Weaknesses:**

1. nit: L213-215: "Given that the sum of two sides ..." I don't really see this as a strong explanation, since the data is actually in a higher dimension space rather than 2D?
2. L238-239: "where only the newly generated data is use ... causing the model’s training to fail to converge" and Model A in the experiment section show that generating new image every batch makes the model couldn't converge. It would be nice if the authors can dive in more in this phenomenon and explain why this would happen. Suppose that the author claimed that GAN-loss would help the generated distribution more aligned with the real image distribution, I don't understand why having new samples every batch would harm it (ignore the computational expense).

-> If possible, authors should provide a more detailed analysis of this phenomenon, perhaps including visualizations of the training dynamics or theoretical justifications for why this occurs despite having GAN loss.

3. It seems like the model needs to be initialized with good weights beforehand, would the method still work if it is randomly initialized? If not, what would be the root cause?

-> If possible, authors should provide an ablation study comparing performance with random initialization versus pretrained initialization, and a discussion of the implications for the method's robustness and generalizability.

4. The `freq` param to update new data seems to be quite important in my opinion, does it need to be tuned properly for this training scheme to work? An ablation study for this would be more convincing.
5. GAN loss is known to be unstable while training, was the training for ASFM affected by it, it yes, how did the authors fix it?
6. Both GAN-loss and reflow is also known for making the model has the property of mode collapse. Do models trained with ASFM have this problem? Authors should also use metrics such as "recall" (https://arxiv.org/abs/1904.06991) to evaluate the model.
7. Is there any reason for the fact that there's no output metrics for ASFM using NFE=6 same as in AFHQ and CelebA-HQ?

**Questions:**

Author should address the Weakness section.

---

> ### Author Response · Authors · 2024-11-22
>
> We would like to sincerely thank the reviewer for the thoughtful and valuable comments on our work. We will address these issues and provide a detailed explanation accordingly, as shown below.
>
> **Q1: Explanation of the straightening effect of Self Online Training.**
>
> **A1:** In fact, both the triangle inequality and the shortest distance between two points being along the straight line hold in any finite-dimensional Euclidean space. In simple terms, when the line connecting two pairs in the training data intersects at a certain point, the vector field learned by the neural network will pass through the vicinity of the intersection and "repair" the two pairs. The repaired line segments will no longer intersect, resulting in a vector field that is straighter.
>
> **Q2: Regarding Model A's failure to converge.**
>
> **A2:** When we conducted the experiments, we found that directly using Online Self Training, with the Training Memory size set to 1, caused the model to fail to converge, as shown by the yellow lines in Figure 6 (a) and (b). Despite the use of the GAN loss to help align the generated data distribution more closely with the real data distribution, we believe that due to the online training nature of ASFM, training with a small Training Memory leads to overfitting on the data in the Training Memory before the memory is updated. This, in turn, degrades the quality of the next batch of generated data in the online training process, causing the model to fail to converge. This is why, when designing the Training Memory size for ASFM on the CIFAR-10 dataset, we chose a memory size that is comparable to the size of the CIFAR-10 training set.
>
> **Q3: Initialization of the network**
>
> **A3:** Even with randomly initialized model parameters, we believe that ASFM can generate data that aligns with the real data distribution, as the discriminator provides information about the real data distribution to the model. However, the random initialization of model parameters presents challenges for data generation during training. Methods and parameters such as random initialization and the weight of the GAN loss need to be explored experimentally. Due to time constraints, we leave this as a direction for future research.
>
> **Q4: Ablation study of parameter *freq*.**
>
> **A4:** We included ablation studies on hyperparameters such as `freq`. Please refer to Appendix E, Figure 10, which demonstrates the effectiveness of our hyperparameter choices.
>
> **Q5: GAN loss may lead to unstable training.**
>
> **A5:** During the training process of ASFM, the FID gradually decreases as the number of training steps increases, demonstrating the stability of the training process, as shown in Figure 6. This may be because, during training, ASFM does not directly input the final sampled results into the discriminator. Instead, it uses the data generated in one step as a correlated representation of the final sampling result to input into the discriminator, thereby reducing the impact of the GAN loss on the training process.
>
> **Q6: If ASFM suffer from model collapse?**
>
> **A6:** We computed the *recall* metrics of ASFM on the CIFAR10 dataset with NFE=1 and NFE=2, and compared them with methods such as 1-rectified flow, as shown in the following table. The results show that ASFM achieves the best recall metrics, which we attribute to the following factors: 1) The training data is generated online, which ensures more comprehensive sampling of the distribution, thereby avoiding situations where certain modes cannot be generated. 2) ASFM’s use of the GAN loss occurs during the one-step generation process within training, where this one-step generation serves as a representation rather than a method for generating data. As a result, ASFM's sensitivity to the GAN loss is limited.
>
> | Methods          | NFE  | recall   |
> | ---------------- | ---- | -------- |
> | 1-rectified flow | 127  | 0.57     |
> | StyleGAN2        | 1    | 0.41     |
> | StyleGAN-XL      | 1    | 0.47     |
> | StyleGAN2 + ADA  | 1    | 0.49     |
> | **ASFM**         | 1    | **0.61** |
> | **ASFM**         | 2    | **0.62** |
>
> **Q7: The results of ASFM with more NFEs on CIFAR10.**
>
> **A7:** Since the flows produced by ASFM are sufficiently straight, results similar to those obtained with more sampling steps can be achieved with fewer steps, as the following table shows.
>
> | solver | NFE  | FID  | IS   |
> | ------ | ---- | ---- | ---- |
> | euler  | 2    | 4.82 | 8.93 |
> | rk45   | 113  | 4.80 | 8.96 |

---

> > ### Comment · Reviewer_7v97 · 2024-11-25
> >
> > > we believe that due to the online training nature of ASFM, training with a small Training Memory leads to overfitting on the data in the Training Memory before the memory is updated
> >
> > Please correct me if I'm wrong, but isn't having a small training memory equal to new samples in every batch? Then in that case diversity should be good and help the model avoiding overfitting?
> >
> > > This may be because, during training, ASFM does not directly input the final sampled results into the discriminator. Instead, it uses the data generated in one step as a correlated representation of the final sampling result to input into the discriminator, thereby reducing the impact of the GAN loss on the training process
> >
> > Could you further explain this? Any backup evidence for this would be better; why does using one-step output reduce the impact of GAN loss? I can see it improves efficiency (skip denoising), but to say it improves stability is far-fetched.
> >
> > >  Since the flows produced by ASFM are sufficiently straight, results similar to those obtained with more sampling steps can be achieved with fewer steps
> >
> > This statement is not aligned with the FID of NFE=1,2,6 in AFHQ-Cat and CelebA-HQ.

---

> > > ### Author Response · Authors · 2024-11-28
> > >
> > > Thanks for the reviewer's feedback. We will further elaborate on these concerns.
> > >
> > > **Q1: The effect of small Training Memory.**
> > >
> > > **A1:** In ASFM, the Training Memory is updated after a certain number of training steps, as described by the parameter `freq` in lines 299-303 of the paper. Specifically, the Training Memory is updated every `freq` training steps. The reason for this design is that updating the memory at every step would result in excessive computational demands and insufficient utilization of the generated data. For instance, when a data pair is added to and later removed from the Training Memory, it undergoes a total of `freq × k` training steps, and the number of times it is selected for training is `1/k × freq × k = freq`. If the memory were updated at every step, a newly generated data pair would only be used for training once.
> > >
> > > In contrast, when the Training Memory size is 1, during a single data update interval, a total of `freq × k = freq` training steps are completed, and the data pair will be used for training `1/k × freq × k = freq` times. On the other hand, when the Training Memory size is `k`, within `freq` training steps, a data pair will be used for training only `1/k × freq` times, which helps prevent overfitting to a single pair of training data within this training interval.
> > >
> > > Since ASFM uses an online training mode, the generated data is immediately used for training. Overfitting to a single training data pair within an update cycle has a greater impact than in offline training, which can affect the model's performance. As shown in Figure 6, when the Training Memory size is 1, the model fails to converge properly.
> > >
> > > **Q2: One-step image generation in training weaken the impact of GAN loss.**
> > >
> > > **A2:** We apologize for any confusion caused by our previous explanation. What we intended to convey is that the instability in GAN training often arises from the mismatch in the training speeds of the generator and the discriminator. In the early stages of training, the generator produces poor results, and the discriminator becomes strong quickly, leaving the generator with ineffective gradient signals. However, ASFM initializes the model parameters with a 1-rectified flow, which, although producing suboptimal results with one-step image generation, is sufficient to reflect part of the final generated information. This helps reduce the training instability caused by the mismatch in training speeds between the generator and discriminator in the early stages.
> > >
> > > **Q3: The results of ASFM with more NFEs on AFHQ and CelebA-HQ.**
> > >
> > > The flows generated by ASFM are sufficiently straight, which means that with very few sampling steps, a slight increase in the number of steps can lead to an improvement in generation quality. However, with a larger number of sampling steps, similar generation quality can be achieved with fewer steps. As shown in the table below, on the AFHQ and CelebA-HQ datasets, the image generation quality increases as the number of sampling steps increases when the sampling steps for AFSM are 1, 2, and 6. Moreover, when the sampling steps are 100, the image generation quality is similar to the result obtained with 20 sampling steps on both AFHQ-Cat and CelebA-HQ.
> > >
> > > | Sampling steps (Euler solver) | FID (AFHQ) | FID (CelebA-HQ) |
> > > | ----------------------------- | ---------- | --------------- |
> > > | 1                             | 44.0       | 79.0            |
> > > | 2                             | 22.9       | 22.2            |
> > > | 6                             | 14.9       | 8.15            |
> > > | 20                            | 11.7       | 4.46            |
> > > | 100                           | 10.2       | 3.80            |

---

### Official Review · Reviewer_5kSX · 2024-11-08

**Soundness:** 3
**Presentation:** 3
**Contribution:** 3
**Rating:** 6
**Confidence:** 3

**Summary:**

The paper introduces Adversarial Self Flow Matching (ASFM), a method for few-step image generation using straight flows in Ordinary Differential Equation (ODE)-based generative models. ASFM includes two components: Online Self Training, which straightens flows by constructing a conditional vector field with paired data, and Adversarial Training, which aligns generated data with real data to reduce cumulative errors.

**Strengths:**

Few-Step Generation: ASFM enables high-quality image generation in just a few steps, which is more efficient compared to traditional methods that require a large number of steps.

Improved Efficiency: By leveraging straight flows, ASFM reduces the number of function evaluations (NFE) needed, significantly decreasing the computational cost and time.

Zero-Shot Capabilities: ASFM retains the zero-shot capabilities of ODE-based generative models, allowing for tasks such as image editing without the need for paired data.

**Weaknesses:**

Complexity in Training: ASFM's two-component training methodology, including Online Self Training and Adversarial Training, may introduce additional complexity in the training process compared to simpler generative models.

Generalization to Other Tasks: While ASFM shows promise for image generation, its extension to more complex tasks like text-to-image generation or large-scale datasets may require further enhancements and research.

**Questions:**

1. How does ASFM balance the trade-off between the quality of generated images and the number of function evaluations (NFE)?

2. Can ASFM be extended to conditional generation tasks, such as text-to-image synthesis, and if so, what challenges might arise?

---

> ### Author Response · Authors · 2024-11-22
>
> We would like to sincerely thank the reviewer for the positive and thoughtful comments on our work.  We address these questions and provide a detailed explanation as described in the following.
>
> **Q1: The trade-off between image quality and NFE?**
>
> **A1:** Since the flows produced by ASFM are sufficiently straight, results similar to those obtained with more sampling steps can be achieved with fewer steps. For example, on the CIFAR-10 dataset, the generation results using the Euler solver with two sampling steps and the results using the RK45 solver are shown below. Although their NFEs (Number of Function Evaluations) differ significantly, the generation quality is quite similar.
>
> | solver | NFE  | FID  | IS   |
> | ------ | ---- | ---- | ---- |
> | euler  | 2    | 4.82 | 8.93 |
> | rk45   | 113  | 4.80 | 8.96 |
>
> In the case of a few sampling steps, increasing the number of steps can improve the generation quality. As shown in Table 1, when using the Euler solver in ASFM, increasing the sampling steps from 1 to 2 leads to an improvement in sampling performance (FID decreases and IS increases).
>
> **Q2: Generalization to other tasks.**
>
> **A2:** Extending ASFM to conditional generation tasks such as text-to-image generation, is a highly worthwhile research problem. We plan to address this issue in our future work.

---

### Note · Authors · 2025-01-31

I have read and agree with the venue's withdrawal policy on behalf of myself and my co-authors.